# A Current Synopsis of the Emerging Role of Extracellular Vesicles and Micro-RNAs in Pancreatic Cancer: A Forward-Looking Plan for Diagnosis and Treatment

**DOI:** 10.3390/ijms25063406

**Published:** 2024-03-17

**Authors:** Eleni Myrto Trifylli, Anastasios G. Kriebardis, Evangelos Koustas, Nikolaos Papadopoulos, Sotirios P. Fortis, Vassilis L. Tzounakas, Alkmini T. Anastasiadi, Panagiotis Sarantis, Sofia Vasileiadi, Ariadne Tsagarakis, Georgios Aloizos, Spilios Manolakopoulos, Melanie Deutsch

**Affiliations:** 1Laboratory of Reliability and Quality Control in Laboratory Hematology (HemQcR), Department of Biomedical Sciences, Section of Medical Laboratories, School of Health & Caring Sciences, University of West Attica (UniWA), Ag. Spyridonos Str., 12243 Egaleo, Greece; trif.lena@gmail.com (E.M.T.); sfortis@uniwa.gr (S.P.F.); 2First Department of Internal Medicine, 417 Army Share Fund Hospital, 11521 Athens, Greece; aloizosgio@yahoo.gr; 3GI-Liver Unit, 2nd Department of Internal Medicine National and Kapodistrian University of Athens, General Hospital of Athens “Hippocratio”, 114 Vas Sofias, 11527 Athens, Greece; vasileiadisofia@gmail.com (S.V.); smanolak@med.uoa.gr (S.M.); meladeut@gmail.com (M.D.); 4Oncology Department, General Hospital Evangelismos, 10676 Athens, Greece; vang.koustas@gmail.com; 5Department of Biological Chemistry, Medical School, National and Kapodistrian University of Athens, 11527 Athens, Greece; panayotissarantis@gmail.com; 6Second Department of Internal Medicine, 401 General Military Hospital, 11527 Athens, Greece; nipapmed@gmail.com; 7Department of Biochemistry, School of Medicine, University of Patras, 26504 Patras, Greece; aanastasiadi@upatras.gr (A.T.A.); vtzounakas@upatras.gr (V.L.T.); 8Beth Israel Deaconess Medical Center, Harvard Medical School, Boston, MA 02215, USA; a.tsagarakis96@gmail.com

**Keywords:** extracellular vesicles, pancreatic cancer, biomarkers, drug delivery, exosomes, microvesicles, therapy, micro-RNAs, non-coding RNA

## Abstract

Pancreatic ductal adenocarcinoma (PDAC) is one of the deadliest malignancies worldwide, while it persists as the fourth most prevalent cause of cancer-related death in the United States of America. Although there are several novel therapeutic strategies for the approach of this intensely aggressive tumor, it remains a clinical challenge, as it is hard to identify in early stages, due to its asymptomatic course. A diagnosis is usually established when the disease is already in its late stages, while its chemoresistance constitutes an obstacle to the optimal management of this malignancy. The discovery of novel diagnostic and therapeutic tools is considered a necessity for this tumor, due to its low survival rates and treatment failures. One of the most extensively investigated potential diagnostic and therapeutic modalities is extracellular vesicles (EVs). These vesicles constitute nanosized double-lipid membraned particles that are characterized by a high heterogeneity that emerges from their distinct biogenesis route, their multi-variable sizes, and the particular cargoes that are embedded into these particles. Their pivotal role in cell-to-cell communication via their cargo and their implication in the pathophysiology of several diseases, including pancreatic cancer, opens new horizons in the management of this malignancy. Meanwhile, the interplay between pancreatic carcinogenesis and short non-coding RNA molecules (micro-RNAs or miRs) is in the spotlight of current studies, as they can have either a role as tumor suppressors or promoters. The deregulation of both of the aforementioned molecules leads to several aberrations in the function of pancreatic cells, leading to carcinogenesis. In this review, we will explore the role of extracellular vesicles and miRNAs in pancreatic cancer, as well as their potent utilization as diagnostic and therapeutic tools.

## 1. Introduction

Pancreatic ductal adenocarcinoma (PDAC) is characterized by an intensely aggressive behavior, while a large number of cancer-related deaths are attributed to this malignancy in the USA [1], constituting the fourth most common etiology and eventually will be the most frequently diagnosed cancer by 2030 [2]. PDAC presents a higher incidence in recent decades, based on the worldwide epidemiologic statistical data, which is more likely attributed to the higher incidence of metabolic syndrome and obesity [3]. The main challenge for clinicians remains the detection of PDAC in its early stages because it is relatively asymptomatic. However, the first symptomatology in PDAC patients is attributed to its highly aggressive course, including the early metastatic tumor progression [4]. 

Several significant risk factors have been identified in PDAC patients, with the foremost being chronic pancreatitis (CP), which significantly increases (7.2 times higher) the risk of PDAC. Some other well-studied factors include obesity, diabetes mellitus, excessive alcohol, and cigarette consumption, as well as industrial chemical exposure [5]. Additionally, another underestimated risk factor is periodontal disease and poor oral hygiene, which is closely related to chronic inflammation and bacterial translocation, implying the crucial role of microbiome in disease development [6]. Meanwhile, Systemic Lupus Erythematosus (SLE) also constitutes a risk factor for PDAC development based on the meta-analytic data [7], while patients who present another non-modifiable risk factor such as Peutz–Jeghers syndrome, related to *STK11* mutation, have a significantly higher risk (132 times higher) of developing it [8]. 

Focusing on the molecular background of this malignancy, several gene mutations are documented including *KRAS*, *BRCA1*, *BRCA2*, *TP53*, *SMAD4*, *CTFR*, *CDKN2A*, *MLH1*, *SPINK1*, *ATM*, and *PALB2*, as well as *PRSS1*, *MSH2*, and *MSH6.* In the cases of familial pancreatic cancer (FPC), the most well-documented genetic alterations include *SPINK1*, *CFTR*, and *PRSS1* mutations, while *CDKN2A* and *BRCA2* are also highly identified in several other hereditary genetic conditions [9].

Based on the aforementioned analysis, the discovery of novel diagnostic tools and therapeutic modalities is considered an emergency to overcome the diagnostic difficulties, the chemoresistance, and the high post-surgical reoccurrence in almost 1/4 of the operated patients [10,11]. Meanwhile, current immunotherapy and chemotherapeutic strategies including immune checkpoint inhibitors (ICIs) and (FOLFIRINOX)-gemcitabine (GEM)—nabpaxlitaxel—are considered inefficacious in most PDAC cases [11]. 

At the center of scientific research is the pivotal role of extracellular vesicles (EVs) and the non-coding micro-RNAs (miRNAs) in PDAC progression, diagnosis, and their potent utilization as therapeutic tools [12,13]. The former molecules are membranous nanoparticles that contain several cargoes, being released by a diverse range of cells [12]. These nanoparticles have a fundamental role in intercellular communication between the parental cell and the recipients, under normal or pathological conditions, like pancreatic carcinogenesis. There are a wide range of cargoes within these vesicles, including protein molecules, receptors, DNA molecules, RNA coding or non-coding sequences, and lipids, as well as autophagosomes. Focusing on non-coding RNA molecules, short, non-coding micro-RNAs (miRNAs) are closely implicated in significant cell functions, including cell proliferation, differentiation, and apoptosis, as well as in major signal transduction and metabolic pathways, although they are not involved in protein coding [14]. The uptake of these EV-containing miRNAs by the recipient cells can lead to the alteration of their functional state on several levels, as previously mentioned, leading to disease development and progression, including PDAC [15]. However, there are also some other non-coding RNA molecules, such as long non-coding RNAs (lncRNAs) and circular RNAs, which also have a pivotal biological role in carcinogenesis. The former can bind with RNA, DNA molecules, and several proteins, having the so-called role of “molecular sink”, while the latter can act as a “sponge” for miRNAs, protecting mRNA translation from the silencing effect of miRNAs, leading to the enhancement of several gene expressions, as well as performing a similar role with RNA-binding proteins [16,17].

Furthermore, it is crucial to underline the dual role of miRNAs in pancreatic carcinogenesis because they either act as promoters, leading to cancer progression, metastasis, and resistance to chemotherapeutic agents, or as suppressors. Shedding light on the EV-miRNA expression profiles of pancreatic tumors, novel diagnostic, prognostic, and predictive biomarkers, as well as new therapeutic targets could be developed [18]. In this review, we will shed light on the interplay between EVs and miRNAs in PDAC development and progression, as well as in their potential utilization as diagnostic and therapeutic tools.

## 2. A Brief Analysis of the EV and miRNA Biogenesis Mechanism 

### 2.1. EV Biogenesis 

EVs have a crucial role in cell–cell communication and their implication in disease progression, including carcinogenesis. They have been the main interest of the scientific community in recent years for their potential use not only as diagnostic tools but also as therapeutic targets or drug vectors [19]. Understanding their biogenetic mechanisms is considered necessary for their utilization in anti-neoplastic diagnostic and therapeutic strategies. These vesicles are quite heterogeneous, which is mainly attributed to the distinct biogenetic pathways, their various sizes, as well as their diverse cargoes. Based on their diameter, they are divided into the following categories: (i) apoptotic bodies (>1000 nm), (ii) microvesicles (150 nm–1000 nm), and (iii) exosomes (40–150 nm). The first entity arises from the cell apoptosis pathway, the second is derived from the outward blebbing of the cell membrane, and the latter arises from the inward budding of the membrane [20]. 

The biogenesis of apoptotic bodies starts with the initiation of cell apoptosis, which requires chromatin condensation, nuclear shattering, and cell shrinking, with the concomitant formation of blebs, which constitute the so-called microtubule spikes, as well as the apoptopodia. At the end of this procedure, the apoptotic vesicles are formed, after the segmentation of the apoptotic bodies [21].

Furthermore, the microvesicles (or ectosomes) have a significant role in several manners, as they carry a wide variety of selected cargo such as integrins, major histocompatibility Complex-I (MHC-I), as well as nucleic acids, and plasmid DNA. Meanwhile, they also contribute to significant signaling pathways, or alter the extracellular space, aiming for several effects like cell invasion. There are some requirements for the blebbing (outward budding) of the cell membrane, such as the transport of TSG101 from the endosomal membrane, which is a late endosomal protein that interacts with the arrestin domain-containing protein-1 (ARRDC1), aiming the alteration of the membrane’s curve for allowing the pinching of the microvesicles towards the extracellular milieu, with the produced vesicles expressing the two aforementioned proteins [22]. Cargo-recruitment is mediated through vertical trafficking towards the plasma membrane in this biogenetic mechanism, while SNARE proteins contribute to the cargo trafficking under hypoxia [23]. 

Moreover, exosome biogenesis has two main pathways based on the dependence on the endosomal sorting complex (ESCRT) contribution in cargo trafficking [24]. Starting with the non-dependent ESCRT pathway, the cargo recruitment starts from different sites than the cell membrane, such as the cytosol or trans-Golgi complex [25]. On the other route that ESCRT machinery is required, the starting point is the internalization of protein molecules or receptors of the cell membrane, which via an inward budding form vesicles that separate from the membrane. These protein molecules, which are tagged via ubiquitin are identified by the components of the ESCRT-0 complex, including Hse1 and Vps27. The Vps27 binds on the endosomal membrane via a lipid (phosphatidylinositol 3-phosphate), followed by the recruitment of all this complex to the endosome. The aforementioned component of ESCRT-0 recruits the ESCRT-I complex via its binding with the Vps23 component of the latter. Similarly, the ESCRT-I complex via the binding between its component Vp28 and the component of ESCRT-II (Vps36), recruits the latter. Bro1/ALIX complex, which are accessory proteins of ESCRT machinery, has an important role in the pathway, as it removes the ubiquitin tag from the proteins, which were going to undergo lysosomal degradation, before the formation of multivesicular bodies (MVBs) [26]. Likewise, ESCRT-II recruits ESCRT-III via the binding between their subunits Vps25 and Vps20, respectively. Meanwhile, several intraluminal vesicles (ILVs) arise from the late endosomal membrane inside the lumen of the MBVs, with the end of MBV’s generation being regulated by the Vta1-Vps4 complex. The last complex of ESCRT machinery is required for the separation of the vesicles from the membrane, while the Bro1/ALIX complex stabilizes it via binding on its Snf7 subunit. Last but not least, once MVBs are formed, they are transferred towards the lysosomes for degradation, or they are fused with plasma membrane for the exocytosis of exosomes, under the effect of SNARE proteins, which include the VAMP7, Ykt6, and Syntaxin1A proteins. However, MVBs can merge with autophagosomes from the autophagy pathway, giving rise to amphisomes, which are further integrated with the cell membrane for the release of exosomes in the extracellular space or sent to lysosomes for degradation [27]. In Figure 1, we present the routes of EV biogenesis and the ESCRT-dependent exosome biogenesis.

### 2.2. MiRNA Biogenesis

MiRNAs constitute short (19–25 nucleotides), non-coding sequences of RNA molecules, which have a core function in the regulation of genome expression. These sequences have a key biological role in several cell functions including apoptosis, autophagy, and hematopoiesis, while they are also implicated in major metabolic and signaling pathways [28]. Their research remains in the spotlight of many studies about carcinogenesis, as they have a dual role as tumor promoters, the so-called oncomiRs, or suppressors. The pathway of miRNA biogenesis starts with the transcription of the miRNA coding genes by RNA polymerase II, which constitutes a procedure that gives rise to primary miRNA (pri-miRNA) and is located within the cell nucleus. The next step of the pathway requires the cleavage of pri-miRNA (over 1000 nucleotides) by DiGeorge Syndrome Critical Region 8 (DGCR8)–Drosha (DGCR8–Drosha) complex, which constitutes a ribonuclease complex. Drosha constitutes a ribonuclease III and together with its cofactor DGCR8 cleaves the stem-loops of pri-miRNA, giving rise to precursor miRNAs (pre-miRNA) [29]. The latter is then exported into the cytoplasm via the Exportin 5 and RAS-related nuclear protein–guanosine-5′-triphosphate (Ran-GTP) complex, which is part of the nucleocytoplasmic transport system. Exportin 5 binds the nuclear pre-miRNA sequence and exports it through the nuclear pore complex, while Ran-GTP ensures the direction of this relocation. The subsequent step of the pathway requires the cleavage of cytoplasmic pre-miRNA by the Dicer–TRBP complex [30]. The enzymatic action of this complex is crucial for the RNA Interference (RNAi) Pathway. More particularly, Dicer cleaves the pre-miRNA into shorter fragments, while TRBP, which is an RNA-binding protein (RBP), enables their loading on RNA-induced silencing complex (RISC) [30]. Duplex miRNAs are the products of the removal of the terminal loop of pre-miRNA molecules by Dicer-TRBP, which are further loaded on the RISC complex that is composed of Argonaute protein and has a catalytic site, the so-called Argonaute RISC Catalytic Component 2 (Ago2) that separates the duplex miRNA into two strands the mature and the passenger. Argonaute has a dual role as a binding site for the leading or active strand (mature miRNA) for the formation of the miRNARISC (miRISC) complex, as well as a guide towards its partially complementary targeted messenger RNAs (mRNAs) [31]. The other strand of the duplex miRNA that is not bound on RISC is the passenger strand, which is further degraded. Then, the miRISC binds on the 3′untranslated region of the targeted mRNA strand, for which miRNA is partially complementary (the seed sequence), which might lead to the silencing and the translational repression of the mRNA, or its cleavage. The RISC complex has a crucial role in gene expression, as it induces gene silencing in several ways, including translational repression and mRNA degradation. In the former condition, mRNA is not further translated into a specific protein, while in the latter, it is degraded [32]. However, there is another alternative, non-canonical pathway of miRNA biogenesis, which does not require the enzymatic action of the DGCR8-Drosha complex for the cleavage of pri-miRNA and the one of Dicer, while there are also differences in the seed regions [33,34]. In Figure 2, there is a schematic representation of the miRNA biogenesis and its implication in gene silencing. 

## 3. An Overview of the Implication of miRNAs and Other Non-Coding RNAs in PDAC

As was demonstrated in the previous chapter of miRNA biogenesis, the mRNA translation can be significantly influenced by miRNAs. MiRNAs can closely regulate the protein-encoding via repressing or silencing the translation or via mRNA degradation, as they can bind on specific target mRNA molecules, which are partially complementary. The expression levels of miRNA have a crucial role in carcinogenesis, as they can induce or suppress tumorigenesis. Deep knowledge of the miRNA tumor profiles including PDAC, could lead to the development of powerful anti-cancer therapeutic tools. There are several alterations in the expression levels of miRNAs in PDAC, which can facilitate its diagnosis, and can be utilized as targets for tumor suppression via inhibiting the oncomiRs or enhancing the expression of tumor-suppressive miRNAs [35]. 

### 3.1. OncomiRs 

►**miR-21**: This miRNA is highly expressed in serum or tissue biopsies. The expression levels of miR-21 are closely related to the regulation of tumor-suppressor genes that are implicated in pivotal cell functions and pathways, such as apoptosis and Ras-Raf-MEK-ERK pathways or/and epidermal growth factor receptor (EGFR), or/and PI3K/AKT signaling pathways, respectively [36,37]. More particularly, miR-21 overexpression induces the growth and proliferation of pancreatic cancer cells, while it concomitantly inhibits their apoptosis, resulting in a decontrolled cell cycle [36,37]. ►**miR 186**: The levels of miR-186 are also highly found in PDAC, which also induces proliferation and metastasis via targeting the Nuclear Receptor Subfamily 5 Group A Member 2 (NR5A2) gene (encodes the transcription factor NR5A2), leading to several deregulations in gene expression [38].►**miR-17-5p**: Its overexpressed levels are closely implicated in the cell cycle deregulation, via interrupting the expression of RBL2/E2F4 repressing complexes [39].►**miR-196b** suppresses the apoptotic mechanism via targeting CADM1 [40]. ►**miR-18a**, which is a member of the oncogenic miR-17-92 cluster, is highly expressed in PDAC, whereas its levels are significantly reduced after surgical treatment. This miRNA is highly implicated in proliferation and MYC-induced transcriptional activation [41]. ►**miR-191** is implicated in the modification of the extracellular matrix and the promotion of distant tumor cell dissemination [42].►**miR-29a** and **miR-221** are also implicated in PDAC progression, promoting invasion and metastasis [43,44,45]. ►**miR-301a-3p** and **miR-374** also have an oncogenic role in PDAC via inducing migration and increasing the invasiveness of pancreatic cancer cells, with the former targeting SMAD4 expression [46], whereas the latter does so via deregulating Secernin 1 (SRCIN1), leading to its low expression, and to EMT and PDAC progression [47]. ►**miR-1469-5p** is correlated to the over-proliferation of PDAC cells, via interacting with the NDRG1/NF-κB/E-cadherin pathway [48]. ►**miR-205** is implicated with the Wnt signaling pathway, increasing the proliferation via targeting the suppressive gene that encodes Adenomatous polyposis coli (APC) [49]. ►**miR-10b:** Its overexpression is closely involved in PDAC invasive behavior and progression via inhibiting TIP30 expression and promoting EGF and TGF-β effects, leading to a generally poor prognosis [50].

In Table 1, we demonstrate a summary of the oncogenic effect of several miRNAs in PDAC.

### 3.2. Τumor-Suppressive miRNAs in PDAC

Several studies have demonstrated a wide variety of tumor-suppressive miRNAs in PDAC; however, their levels are commonly decreased, resulting in tumor initiation and progression. 

►**miR-506:** Enhanced levels of miR-506 were closely related to the suppression of pancreatic cancer cell growth and chemosensitivity; however, it is usually detected in low levels. More particularly, its downregulated levels are correlated to chemoresistance via sphingosine kinase 1 (SPHK1)/Akt/NF-κB signaling [51]. ►**miR-34** also induces tumor suppression, when its levels are reinstated, leading to the inhibition of the cancer stem cells [52]. ►**miR-142:** It is reported that miR-142 restored levels are closely related to the limitation of the tumor invasive behavior and growth via regulating the expression of hypoxia-inducible factor 1-alpha (HIF-1a), which is a transcription factor. However, this miRNA is usually presented in low levels in PDAC cases [53]. ►**miR-216b:** Increased expression of miR-216b has tumor-suppressive effects via its implication in the expression of translationally controlled 1 tumor proteins (TC1TP) as well as via KRAS inhibition [54]. ►**miR-30c:** The enhancement of its levels has also shown PDAC suppression via targeting TWF1, which has a key role in cell cycle regulation (G1) and in programmed cell death [55]. ►**miR-143-3p:** Increased miR-143-3p expression levels lead to the suppression of MERK/ERK signaling pathway and limit the pancreatic cell dysplasia [56]. ►**miR-519-3d** suppresses hypoxia-related carcinogenesis by regulating programmed death ligand 1 (PD-L1) [57]. ►**miR-1181** induces the suppression of STAT3, limiting the invasiveness and progression of PDAC [58].►**miR-375** induces increased PDAC cell apoptosis, as well as limits the lymphatic spread and distant metastasis [59]. ►**miR-455-3p** and **miR-135a** are also PDAC-suppressive, with the former inducing increased apoptosis of cancer cells, EMT inhibition, and regulation of TAZ expression, whereas the latter is implicated in the expression of Bmi1 [60,61].►**miR-340:** The enhancement of its expression levels also limits tumor progression and proliferation via targeting the expression Bicaudal-D2 (BICD2) [62].►**miR-203a-3p** also reduces PDAC progression and invasion via its implication in fibroblast growth factor 2 (FGF2) expression, leading to EMT limitation [63]. 

In Table 2, we demonstrate a summary of the oncogenic effect of several miRNAs in PDAC.

### 3.3. A Brief Review of the Role of Other Non-Coding RNAs in PDAC

Long non-coding RNAs (lncRNAs) constitute another type of non-coding, but biologically active RNA molecules that can also alter the functionality of the other cells, at the level of gene expression, such as by modifying chromatin, by regulating X-chromosomal functions, and by catalyzing histone methylation and acetylation at the epigenetic level [64]. These lncRNAs are longer in size compared to miRNAs and they also have a pivotal role in the expression of genetic information. Aberrations in their expression levels are described in several diseases, including PDAC, while they have a crucial role in carcinogenesis via interacting with other RNA or DNA molecules and/or with proteins, by promoting tumor aggressivity, progression, neoangiogenesis, migration, and chemoresistance [65]. However, despite the crucial effect of lncRNAs as “molecular sinks”, there are also the circular RNAs (circRNAs), which have a pivotal biological role in carcinogenesis. The latter can act as a “sponge” for miRNAs, protecting mRNA translation from the silencing or degradative effect of miRNAs, while they can interact with RNA-binding proteins [66]. All the aforementioned phenomena are included in the competitive endogenous RNA (ceRNA) theory by which circ- or lncRNAs can competitively share miRNA binding sites, which opens up new therapeutic horizons for PDAC [67]. There are some major PDAC-related lncRNAs or circRNAs which can be potentially utilized as diagnostic or therapeutic tools. 

#### 3.3.1. Oncogenic lncRNAs Implicated in PDAC

►**HOTTIP** is correlated with chemoresistance, which is widely expressed in PDAC tissues, compared to normal non-malignant ones, promoting PDAC progression and GEM-resistance [68]. On the other hand, its suppression could be a future druggable target, as it is reported that the knockdown of this lncRNA suppresses tumor growth and cells sensitize to GEM [68].►**PVT1, HOTTIP, and HOTAIR:** Based on the current study that was conducted by Jiang XY et al. (2023) several lncRNAs are implicated in PDAC proliferation, migration, and chemoresistance, which can be utilized also as druggable targets and diagnostic biomarkers, including plasmacytoma variant translocation 1 (PVT1), HOTTIP, and HOTAIR [69]. These molecules are closely implicated in PDAC progression via inducing EMT, interacting with signaling pathways, and binding several miRNAs. More particularly, PVT1 is closely related to PDAC progression and GEM-resistance, as it acts as a “sponge” for the miR-619-5p that induces autophagy activation and is implicated in Pygopus2 increased expression. HOTAIR is considered quite oncogenic, with its expression levels being negatively correlated with PDAC prognosis, overall survival, and lymphatic dissemination [69].►**GSTM3TV2** contributes in GEM resistance by increasing the expression levels of L-type amino acid transporter 2 (LAT2) and oxidized low-density lipoprotein receptor 1 (OLR1) by sponging let-7 [70]. ►**XLOC_006390, HOTTIP-005, and RP11-567G11.1** are notably increased in PDAC samples, implying its potential as diagnostic tools in PDAC [71].►**linc00511** is highly expressed in PDAC tissues and it is closely associated with worrisome prognosis, PDAC progression, and neoangiogenesis by inducing upregulation of VEGFA. The aforementioned phenomenon is mediated via competing the binding activity of has-miR-29b-3p towards its partially complementary mRNA, which encodes VEGFA protein. Linc00511 constitutes a novel prognostic biomarker, as well as a possible druggable target via its knockdown [72]. ►**MALAT-1, AFAP1-AS1, AF339813, and H19** are tumor-promoting lncRNAs, which are overexpressed in PDAC cells and patient samples [73]. More specifically, lncRNA AF339813 upregulates NFUF2 mRNA translation in PDAC cell lines, while its knockdown could serve as a future druggable target [73,74]. Moreover, MALAT1 induces activation of the autophagy pathway. PDAC constitutes a malignancy that presents overregulation of autophagy that needs to be suppressed, compared to other malignancies, in which autophagy inhibition could lead to oncogenesis [35,73,75,76]. Furthermore, AFAP1-AS1 promotes PDAC growth and invasion by upregulating the IGF1R oncogene via miR-133a sequestration, which regulates its translation [77]. LncRNA H19 acts as a sponge of several specific miRNAs such as let-7b, miR-107, miR-874, miR-130a, and miR-200, as well as miR-675 and miR-194, leading to the deregulation of the expression levels of several proteins and signaling pathway, while its expression levels are higher in PDAC patients samples, like saliva [78,79]. Meanwhile, its knockdown demonstrated tumor suppression in xenografts. In addition, the lncRNA regulator of reprogramming (ROR) competes with miR-145 and induces its sponging [80], leading to the downregulation of *Nanog* expression in Capan-1 and BxPC-3 cell lines, which is a phenomenon that promotes pancreatic cell proliferation [80].►**ENST00000480739** is another lncRNA that is negatively associated with the PDAC patient prognosis, by up-regulating osteosarcoma amplified-9 (OS-9) that interacts with hypoxia-inducible factor 1 (HIF-1), leading to hypoxia-related adaption, invasion, and metastatic dissemination [81].►**NUTF2P3-001** is induced by hypoxia, facilitates Panc-1 and BXPC-3 cell line proliferation in Panc-1, and is correlated with KRAS overexpression [82].

#### 3.3.2. Tumor-Suppressive lncRNAs

►**GAS5** is downregulated in PDAC, whereas its high levels suppress PDAC cell proliferation [83].►**BC008363** is usually downregulated in PDAC; however, when it is overexpressed, it is notably correlated to better survival levels, implying its utilization as a prognostic PDAC biomarker [84]. 

In addition, the identification of lncRNAs that are implicated in the regulation of N1-methyladenine (m1A), N6-methyladenine (m6A), and 5-methylcytosine (m5C), could be utilized as novel prognostic and early diagnostic biomarkers, as well as monitoring tools for immunotherapy response in PDAC patients [85]. 

#### 3.3.3. CircRNAs Implicated in PDAC

The role of hsa_circ_0007367 has been demonstrated in the study by Zhang Q et al., in which hsa_circ_0007367 acts as a sponge for miR-6820-3p, which promotes PDAC progression via upregulating the expression of YAP1 [86]. Moreover, it has been demonstrated that circRNAs interact with signaling pathways such as MAP2K2, BRAF, PI3K/AKT, and WNT/β-catenin signaling [87]. 

Additionally, in the recent study by Xu C et al. (2023) regarding the identification of novel early circRNA-based diagnostic tools, 10 circRNAs were demonstrated as potential PDAC diagnostic tools, after the genome-wide profiling of two databases from the Gene Expression Omnibus. More specifically, the next phases of the research, included the identification of these circRNAs in PDAC tissues, their validation in plasma in a patient cohort, as well as their performance evaluation in different PDAC stages, among other GI tumors and in correlation with CA19-9 levels, which finally gave rise to a diagnostic panel of five circRNAs, including hsa_circ_ 0060733, _0007367,_0006117, _0007895, and _0006117. 

Meanwhile, the aforementioned panel showed an increased AUC in combination with CA19-9 levels (AUC: 0.94) and effective results in PDAC identification in patients with low CA-19-9 levels (<37 U/mL), while no significant difference was reported in PDAC identification of early or late disease stages. Last but not least, PDAC differentiation by other gastrointestinal tumors demonstrated an AUC of 0.83, which was higher in comparison with the AUC for the other GI tumors [88]. 

## 4. The Effect of PDAC-Derived EV-miRNAs and EV-Proteins

EVs are closely implicated in PDAC progression, chemoresistance, and TME modifications. These vesicles transfer several biomolecules that are significantly implicated in several pivotal signaling pathways and alter the translational level of the recipient cells, leading to PDAC progression, invasion, and metastatic dissemination [89]. MiRNAs have a pivotal role in the PDAC progression as was previously referred to in the previous section. The production of several EVs from PDAC cells and the TME components can significantly alter significant metabolic and signaling pathways, as well as the translational level of the recipient cells [90]. 

### 4.1. Implication of EV-miRNAs in Glucose Homeostasis and PDAC-Related DM

►**Exosomal miR-197-3p, miR-6796-3p, miR-4750-3p, and miR-6763-5p:** These exosomes that are derived from PDAC cells deregulate glucose homeostasis by altering the expression of two major peptides, glucagon-like peptide-1 (GLP-1) and glucose-dependent insulin tropic peptide (GIP), in vitro [91].►**Exosomal miR-125b-5p, miR-450b-3p, miR-666-3p, miR-883b-5p, and miR-540-3p:** These exosomes that are derived from PDAC cells induce insulin resistance in C2C12 myotube cells by interrupting PI3K/Akt/FoxO1 axis [91].

Moreover, based on the fact that a big portion of PDAC patients (≤85%) are diagnosed with high glucose levels or DM, up to 3 years before PDAC diagnostic time, several studies are trying to analyze the potential role of EVs and their cargoes in the PDAC-related DM [92]. The role of PDAC–EVs in the background of insulin resistance and DM was demonstrated in the study by Kim, Y.-g (2023) [93]. More particularly, they reported aberrations in the miRNAs expression levels, in PDAC specimens, which were treated with PDAC-exosomes. 

►**Exosomal hsa-miR-3133, hsa-miR-144-5p, and hsa-miR-3148** were proposed as candidate markers and their potential role in the development of insulin resistance and/or DM-associated PDAC, was also demonstrated, when they were exposed to normal pancreatic islets [93].►**EV-miR-19a** was proven to alter insulin production, via interacting with the expression of the gene for Neurod1 protein, which constitutes an important transcription factor, implicated in the β-cells development and differentiation. The aberrations in NeuroD1 are closely implicated with DM and β-cells regulation, leading to reduced insulin secretion in DM-associated PDAC [94].

### 4.2. Implication of EV-miRNAs in PDAC Progression and TME Modification

Meanwhile, some other EV-miRNAs derive from PDAC and promote cancer cell proliferation.
►**EV-miR-222**: Its levels are correlated to the stage and size of PDAC [95].►**EV-miR-155**: Its increased levels are closely associated with suppression of apoptosis in PDAC cells, GEM chemoresistance, as well as induction of EV-miR-155 secretion by the other PDAC cells [96]. ►**EV-miR-125b-5p**: It induces TME alterations and PDAC progression via the activation of the MEK/ERK pathway that leads to PDAC invasion, EMT, and metastatic dissemination [97]. ►**EV-miR-27a**: secreted by PDAC cells, significantly increases angiogenesis through B-cell translocation gene 2 (BTG2), which has anti-proliferative properties, as well as induces PDAC invasion, human microvascular endothelial cells (HMVEC) angiogenesis progression, and metastatic dissemination. The aforementioned phenomenon implies the potential role of miR-27a suppression as a druggable target [98].

### 4.3. ΕV-Proteins Derived from PDAC Cells

The advances in proteomics have opened up new horizons for the identification of the protein cargoes of EVs that can be implicated in PDAC development and progression [99]. There are reported aberrations in the expression profiles of \ EVs-containing proteins derived from PDAC cells, compared to the ones that originate from normal cells.
►**EV-CKAP4:** An interesting observation in surgically treated PDAC patients was the significantly decreased post-operative levels of EV-containing cytoskeleton-associated protein 4 (CKAP4), which were increased before the operation. These EVs are implicated in PDAC progression by interfering with the Wnt signaling pathway [100].►**Circulating EV-O-glycan-binding lectin** were also notably overexpressed in PDAC patients before the surgical treatment, whereas were significantly decreased after pancreatectomy [101].►**EV-β2-microglobulin (B2M):** The phenomenon of tumor escape was significantly correlated with the increased levels of EV-β2-microglobulin (B2M) [102].►**EV-Epidermal Growth Factor Receptor (EGFR), EV-KRAS, and EV-CD44:** Their increased levels have been correlated with pancreatic oncogenesis and PDAC progression [102]. ►**Exosomal-KRAS**: their increased levels can be utilized as a prognostic biomarker, as they are related to shorter overall and progression-free survival [102,103]. ►**EV-Caveolin-1 (CAV1) and EV-Clusterin (CLU)** are closely implicated in the over-proliferation and the impaired apoptosis of pancreatic cancer cells [103]. ►**EV-alpha-3 subunit of integrin (ITGA3):** It is associated with the interaction between PDAC cells and the matrix [102,103,104,105,106].►**EV-Podocalyxin-like protein (PODX)** is associated with cell migration and the invasion into the proximate tissue via cellular protrusions [102,103,104,105,106].►**EVs with Tspan8, Integrins, and CD151** induce stromal changes, promoting cell motility and increasing their invasion and metastatic capacity [107]. 


*EV-proteins implicated in Pre-metastatic niche formation*


Several EVs that are derived from PDAC cells are containing proteins are implicated in the preparation for metastatic dissemination in distant organs, the so-called pre-metastatic niche [108]. 

►**EV-Integrin Beta-5 (ITGΒ5)** influences cell adhesion and intercellular communication between PDAC cells [102,109].►**EV-S100A4** is implicated in cell motility [102,109].►**EV-Annexin A1 (ANXA1)** is implicated in apoptosis and inflammation [102,109].►**EV-tissue factor 3 (F3)** is implicated in immune cell recruitment [102,109].►**EVs containing STAT14, LAMP1, and Lin28B** are implicated in metastasis [102].►**EV-macrophage migration inhibitory factor (MIF)** is implicated in the pre-metastatic niche formation in the liver and finally in the development of the liver metastatic lesions. More particularly, MIF induces the secretion of fibronectin and TGFβ by the hepatic stellate cells and Kupffer cells, respectively. The increased levels of MIF have been closely correlated with the early stages of PDAC, with a concomitant increase in cytokine levels in patients of stage I, who eventually presented liver metastasis [110,111]. 

Last but not least, Kimoto et al. (2023) demonstrated the implication of PDAC-EVs in the formation of a pre-metastatic niche by marking the ascitic-EVs and injecting them in nude mice. Their lungs and liver were histologically analyzed, while they presented distant metastasis, as well as increased permeability in the vessels, compared to mice who received EVs from healthy controls [112]. The above phenomenon was attributed to the EMT modification in the HUVEC cells, as well as to the increased vascular permeability [112]. 


*Poor prognosis and survival, lesion differentiation*
►**EV-Glypican-1 (GPC1):** The increased circulating amount of EV-GPC1, which is attributed to epigenetic alterations in PDAC, is related to poor prognosis and decreased survival. Additionally, its increased levels can differentiate PDAC patients from healthy individuals, or PDAC patients from those who present benign pancreatic diseases, with the latter differentiation being controversial in other recent studies [113]. More specifically, another study demonstrated that glycoprotein 2 (GP2) and GPC1+ EVs are not enough for the proper differentiation between malignant pancreatic lesions and benign ones [114].►**EV-EphA2:** Its increased levels have also been correlated with the tumor stage, although it was proven that they present limited sensitivity in PDAC early stages. Additionally, their levels have been related to the prediction for neoadjuvant treatment response. The increased levels of EV-EphA2 have been related to favorable responses to neoadjuvant therapy, implying their possible use as a monitoring tool for treatment response [115]. ►**EV-Mucin 1 (MUC1) and EV-claudin-1 (CLDN1)** are also reported as exosomes that are closely related to unfavorable and worrisome prognosis [102,116,117]. ►**EVs containing HIST2H2BE, CD151, CLDN4, LGALS3BP, and EpCAM:** Another enlightening study was conducted by Castillo et al., which analyzed the surface protein markers of PDAC exosomes, the so-called “surfaceome”, such as HIST2H2BE, CD151, and CLDN4, as well as LGALS3BP and EpCAM [118]. The authors identified that 73% of exosomes with the selected markers had KRAS mutations, while from the whole sample population, KRAS was detected in 44.1% [118]. ►**EV-zinc transporter (ZIP4):** Its levels have been found upregulated in highly malignant PDAC, compared to moderate ones and healthy controls. It was demonstrated that EV-ZIP4 has an oncogenic potential and it could be utilized in the future as a druggable target [119].►**EV-Adrenomedullin:** it has a pivotal role in DM-related PDAC, while it could be potentially utilized as a marker for β-cell destruction in this malignancy [120].

In Table 3, we demonstrate a summary of the PDAC-derived EV-miRNAs and EV-proteins and their implication in disease progression. 

## 5. The Implication of EVs in the PDAC Microenvironment and Tumor Escape Phenomenon

The tumor microenvironment (TME) plays a pivotal role in PDAC progression, invasion, and neoangiogenesis, as well as migration and metastatic dissemination. As is widely known, TME is a dynamic system composed of cell types lying upon a dense stroma [121]. The aforementioned cellular components interact with each other by releasing several biomolecules, such as cytokines, and EVs with their various cargoes that can potentially intensify the tumor invasiveness, as they promote PDAC progression and expansion. Some of the cellular components of TME are cancer-associated fibroblasts (CAFs), T-regulatory (Treg) cells and tumor-infiltrating lymphocytes (TILs, tumor-associated macrophages (TAMs), B-regulatory cells, and natural killer (NK) cells, as well as several myeloid-derived suppressor cells (MDSCs), pancreatic stellate cells (PSCs), and dendritic and endothelial (ECs) cells. A pivotal manner of cell-to-cell communication is via EV secretion, with these vesicles containing a variety of cargoes. There are three distinct processes of anti-neoplastic immune reaction: (i) the elimination, (ii) the equilibrium, and (iii) tumor immune escape. In the first process, the immune system recognizes and eliminates tumor cells by identifying their neoantigens, while in the second, there is an equilibrium between tumor cell elimination and growth [122]. However, tumor cells escape the anti-neoplastic immunosurveillance and they proliferate. EVs have a significant role in the aforementioned phenomenon, as they act as mediators of several cargoes that can lead to PDAC progression, immune escape, and chemoresistance. It is considered beneficial to identify these EVs, as they can be used as biomarkers, drug vectors, targets for immunotherapy, as well as tumor-associated immune activators [123].

### 5.1. PDAC-Derived EVs That Are Implicated in TME Modification 

Focusing on the PDAC-derived EVs can alter the functionality and the survival of the immune cells, reducing their anti-cancer reaction, leading to their apoptosis, altered immunosurveillance, and eventually to PDAC progression.
*Implication in immune cell functionality*

An obstacle to the anti-neoplastic immune responses can also be induced via EV-related T cell death, as was demonstrated in the study by Shen et al. (2020), as well as via the p38 MAPK pathway and the initiation of Endoplasmic reticulum stress [124]. However, PDAC-EVs also influence the functionality of NK cells.
 ►**EV-Integrins** are correlated with reduced anti-cancer immune response, which is attributed to the decreased expression of INF-γ, CD107a, and TNF-α in NK cells [125]. 

Likewise, the function of mast cells is also altered via PDAC-EVs, through A3 adenosine receptors and induction of MAP and ERK1/2 kinases, which eventually lead to PDAC progression [126].


*Neoangiogenesis, lymphangiogenesis, and metastatic dissemination*
►**EV-ANXA1,** which is accompanied by increased ANXA1 levels has a pivotal pro-angiogenic role in TME, as it significantly modifies the function of endothelial cells and fibroblasts. More particularly, it was demonstrated that ANXA1-EVs activate the formyl peptide receptors (FPRs) and subsequently induce the modification of ECs and fibroblasts, implying the potential role of these EVs as diagnostic and prognostic tools [109]. It is also reported that the aforementioned EVs can induce neoangiogenesis by interacting with ECs and being implicated in the Akt/ERK pathway [127]. ►**EV-miR-27a** promotes lymphatic metastasis and lymphangiogenesis by targeting BTG2 expression [98].►**EV-SUMOylated heterogeneous nuclear ribonucleoprotein A1 (hnENPA1)** promotes lymphatic metastasis and lymphangiogenesis [98].

In addition, it has also been demonstrated in xenografts (mice) that the loss of PRKD1 expression in PDAC cells, induces an increase in F-actin in their plasma membranes, which leads to increased EV secretion and the promotion of PDAC cell’s metastatic dissemination in the lungs [128]. 


*Modification of non-malignant pancreatic cells*


Moreover, PDAC-derived EVs can modify the non-malignant pancreatic cells in the tumor vicinity, as demonstrated in the study by Hinzman et al. (2022). More specifically, they treated non-malignant ECs with PDAC-derived EVs and they observed unfolded protein response (UPR), as well as endoplasmic reticulum (ER) stress in a span of 24 h. The aforementioned phenomenon is attributed to the regulation of the expression of DDIT3, which constitutes a UPR mediator [129]. Meanwhile, they demonstrated the significance of lipid-embedded cargoes in disease progression [129].
 ►**EV-palmitic acid:** These EVs are capable of inducing ER stress in physiological pancreatic cells [129]. 


*Interactions with TAMs and PSCs in TME*
►**EV-Ezrin (EZR)** interacts with TAMs and they can potentially alter the polarity of macrophage and promote tumor metastatic dissemination [130]. This phenomenon can potentially open up new therapeutic chances by targeting the EZR-EVs [130]. Moreover, in the aforementioned study by Chang YT et al. (2020), circulating levels of EZR levels and plasma EZR-EVs in PDAC patients were found higher in comparison to healthy controls [130]. Additionally, they conclude that patients with a higher amount of the aforementioned EVs had a reduced overall survival compared to the group with low levels of EZR.►**EV-miR-155-5p** induces the alteration of macrophages from M1 to M2 immunosuppressive state via interacting with the EHF/Akt/NF-kB pathway [131]. ►**EV-lin28B**: These vesicles induce overexpression of PDGFB in the recipient cells, which leads to metastatic dissemination and pancreatic stellate cells (PSCs) recruitment [132]. ►**Exosomes containing FGD5-AS1**: It was demonstrated that when these EVs are co-cultured with M2 macrophages, they promote tumor progression via the activation of STAT3/NF-κB pathway, which worsens the prognosis in PDAC patients [133].


*Promotion of PDAC invasion, migration, and dissemination*


Moreover, several exosomal miRNAs that are originated by pancreatic cancer cells can significantly alter the TME.

 ►**EV-miR-125b-5p,** which can activate the MEK/ERK pathway, induces PDAC invasion, EMT, as well as metastatic dissemination [97]. It has to be underlined that these EVs can accelerate PDAC development by preventing the targeting of StAR-related lipid transfer protein domain 3 (STARD13), which is a protein family with a crucial role in cancer development and the regulation of inter-organelle cholesterol transportation [90,134]. This phenomenon is derived from the fact that the metabolism of cancer cells is quite accelerated, requiring a high need for cholesterol to fulfill their continuous proliferation and membrane restoration [90,134,135].  ►**EV-LDL receptor-related proteins** activate Yes1-associated transcriptional regulators (YAP) [136]. 

### 5.2. CAFs-Derived EVs


*PDAC progression and chemoresistance*
►**EV-miR-21:** These EVs are correlated with GEM-chemoresistance, a phenomenon that is attributed to their implication in PTEN expression and its inhibition [137,138]. However, the study by Richards et al. (2022) demonstrates the favorable effects of exosome inhibitor GW4869 in vivo and in vitro to restore the expression of PTEN, which is commonly lost in PDAC [138].►**EV-miR-3173-5p:** In the study (ChiCTR2200061320) by Qi et al. (2023), the CAF-derived miRNAs were analyzed, and it was demonstrated that EV-miR-3173-5p promoted chemoresistance to GEM in a xenograft PDAC mouse model via interacting with Acyl-CoA Synthetase Long-Chain Family Member 4 (ACSL4) gene, as well as by inducing suppression of ferroptosis in the recipient PDAC cells, which is a pivotal process of programmed cell death in cases of excessive lipid peroxidation [139]. The aforementioned phenomenon opens up new therapeutic strategies for the management of GEM-resistant tumors via targeting EV-miR-3173-5p [139].►**EV-ANXA6:** In the study by Nigri et al. (2022), it was demonstrated that CAF-derived EVs containing ANXA6-EVs promote PDAC aggressive behavior via the overexpression of CD9 on the surface of these EVs, which induces MAPK pathway activity, increases EMT, and promotes PDAC expansion. Inhibiting these CD9-positive ANXA6-EVs could potentially reduce stromal modification and tumor progression [140]. ►**EV-miR-106b** has been significantly implicated in GEM resistance via its interaction with TP53INP1 in PDAC [141,142]. 

### 5.3. MDSC-Derived Evs

Evs are also implicated in MDSCs increased survival, which has an immunosuppressive and tumor-promoting effect through interacting with the STAT3 pathway, resulting in the upregulation of Chemokine (C-C motif) ligand 5 (CCL5) molecules and (C-C motif) ligand 2 (CCL2) and eventually the further recruitment of tumorigenic immune cells [143].

### 5.4. Total Blood EVs and Leukocyte- and NK-Derived EVs

The analysis of blood-circulating EVs in a multicenter prospective study by *Brocco* et al. demonstrated a higher amount of total blood EVs in PDAC patients, in comparison to healthy individuals, which was independently interrelated with increased progression-free survival and PDAC control rate for patients under chemotherapy treatment [144]. Meanwhile, the levels of blood EVs were significantly decreased in patients with late-stage PDAC, who were receiving chemotherapy. The aforementioned observation implies the potential use of blood EVs as a tool for personalized therapeutic management. Likewise, they observed a higher amount of leukocyte-derived EVs in patients with unresectable or borderline resectable PDAC, with their level being an independent factor for increased overall survival [144]. 

### 5.5. Pancreatic-Stellate EVs

►**EV-miR-451a or miR-21-5p:** The study of Takikawa et al. demonstrated that PSC-EVs that contain miR-451a or miR-21-5p significantly promote the PDAC progression and expansion via promoting the expression of CCL1 and CCL2 [145]. 

### 5.6. TAMs-Derived EVs 

►**EV-miR-21a-5p** leads to PDAC progression [90]. ►**EV-miR-501-3p:** In the study by Yin et al., EV-miR-501-3p derived from TAMs suppresses the expression of the TGFBR3 tumor-suppressive gene, as well as via the activation of the TGF-β pathway that promotes tumor progression [146,147]. ►**EV-miR-301a-3p** derived by the M2 phenotypic state promotes PDAC progression and metastasis via PTEN/PI3Kγ pathway [148]. 

### 5.7. Stromal EVs and Stromal Modification 

The interaction between the acellular components of PDAC TME and the cellular ones has a pivotal role in tumor progression, invasion, and metastatic dissemination [149]. PDAC stroma is significantly modified via PDAC-EVs, which are implicated in the transformation of fibroblasts to CAFs [149,150]. Stromal EVs have a key role in the enhancement of drug resistance and the failure of immune surveillance, leading to PDAC aggressivity and tumor escape phenomenon [149,150,151,152]. More particularly, EVs from PDAC cells induce monocyte-related IL-10 overexpression, which activates the PI3K/Akt/mTOR pathway, while they also suppress the NOTCH signaling pathway, which has a pivotal role in cell differentiation, development, and EMT. The overregulation of this pathway leads to uncontrolled cell proliferation, decreased apoptosis, and carcinogenesis [152]. 

## 6. The Effects of EVs in PDAC-Associated Thrombosis

►**EV-tissue factor (TF)**: TF constitutes a pro-coagulant protein that can be contained in the EVs that are derived from several types of host cells under pathological conditions, including cancer [153]. It has been demonstrated that TF-EVs promote thrombosis in PDAC patients, while baseline TF levels at the start point of chemotherapy constitute a predictive factor of cancer-related thromboembolism, as was demonstrated in Japanese PDAC patient cohorts [154]. It was also demonstrated in the study of Kobayashi et al. (2021) that TF levels above or equal to 100 pg/Ml constitute an independent predictive factor for cancer-associated thromboembolism, while it was also proposed that D-dimers and microvesicle tissue factor (MV-TF) could be utilized as biomarkers for venous thromboembolism in PDAC and may facilitate the identification of patients for whom thromboprophylaxis has to be administrated [154,155]. 

Last but not least, Baj-Krzyworzeka M et al. demonstrated the pro-angiogenic activity of EVs originated from human PDAC cell lines (HPC-4) in vitro, as well as in immunodeficient mice in vivo [156]. It was also observed that these EVs induced increased proliferation and migratory behavior of human umbilical vein endothelial cells (HUVEC), as well as induced cytokine production, such as VEGF and IL-8, which constitute pro-angiogenic factors that facilitate neovascularization [156]. 

## 7. EV Secretion and Hypoxia in PDAC

PDAC is characterized by poor vascularization and increased desmoplasia. The PDAC cell survival under hypoxia requires several mechanisms of adaptation, in which EVs have a crucial role. PDAC cell lines, including AspC1 and MiaPaCa, secrete a high amount of EVs under hypoxic conditions. The size of EVs was significantly altered during extreme lack of oxygen, being noticeably smaller in diameter [157]. 

►**Exosomal-circPDK1:** Lin et al. (2022) suggested that the induction of exosomal circPDK1 by HIF1A under hypoxia leads to PDAC cell survival and migration via c-myc stimulation. The induction of c-myc promotes glycolysis in vivo, as well as in vitro by miR-628-3p sponging that leads to Bromodomain and PHD Finger-Containing Transcription Factor (BPTF) deregulated expression, which has a key role in the regulation of gene encoding [158]. On the other hand, the elimination of these EVs leads to a reduction in tumor migration and proliferation, as well as to shorter PDAC cell survival [158]. 

## 8. A Summary of EV-Mediated Chemoresistance 

In this section, we summarize all the types of cells that secrete EVs that can potentially modify the drug sensitivity of PDAC to chemotherapeutic agents, including CAFs, TAMs, PDAC, cancer stem cells, and GIPC-depleted pancreatic cells or pancreatic cells sensitized to GEM. Meanwhile, it has to be underlined that EVs can also interact with the antibodies in the circulation, leading to the limitation of their effectiveness against the target cells and in the escape of PDAC cells from immunosurveillance [159].

►**EVs containing miR-155 or CAT or SOD2** induce GEM resistance when they are added to pancreatic cell cultures [95].►**EV-miR-210**, which is derived from BxR cancer stem cells (CSCs), induces GEM resistance [160]. ►**EV-ATP-binding cassette (ABC) superfamily G member 2 (ABCG2)**, which is derived from GIPC-depleted PANC1 or AsPC-1, induces GEM resistance [161].►**EV-EphA2**, which is derived from PANC1 cells, induces GEM resistance [115,162].►**EV-miR-155**, which is derived from PANC1 and MiaPaCa2, induces GEM resistance [96,162]. Additionally, exosomes with miR-155 have also been closely implicated in GEM metabolism and inactivation, via suppressing the GEM-metabolizing enzyme DCK in pancreatic cancer cells [163].►**EV-MMP14**, which is derived from BxPC3-Gem cells, induces GEM resistance [162].►**EV-SNAIL-mRNA or miR-106b**, which are derived from CAFs, interact with the recipient pancreatic ECs [138,162]. Additionally, CAFs produce a high amount of EVs under exposure to nab-paclitaxel and GEM, with the recipient cells being resistant to GEM [163]. ►**EV-fibronectin and chitinase 3-like-1 (CHI3L1)**, which are derived from TAMs, are closely implicated in the responses to PDAC treatment via inducing PDAC resistance to GEM [164,165]. However, the inhibition of these molecules by pirfenidone and pentoxifylline for FN1 and CHI3L1, respectively, was demonstrated. Their inhibition partially restored the sensitivity of PDAC cells to GEM, implying the potential role of the aforementioned proteins as druggable targets for PDAC adjuvant treatment [165]. 

## 9. EV-Mediated Cachexia 

A major complication of PDAC is cachexia, which is a state of muscle weakening and atrophy that significantly worsens survival. It was observed that one of the mechanisms of cachexia development is via the induction of EV release that is correlated to muscle loss, as was proposed by Yang J et al. (2019). This phenomenon was mediated via the activation of the CREB-regulated expression of RAB27B, which is promoted by ZIP4 (protein member of the zinc transporters) in an animal model (mice) [119,166,167]. In Table 4, we demonstrate a summary of EVs from several cell origins, their recipient cells, and their effects in PDAC. 

## 10. EVs as Diagnostic Tools in PDAC

The identification of non-invasive biomarkers for PDAC has garnered considerable research interest as this malignancy is quite associated with a high mortality rate. Exosomes have a key role in intercellular communication in pancreatic diseases, including PDAC, which can be utilized as diagnostic, predictive, prognostic, screening, and monitoring tools. These vesicles have a pivotal role in several functions of pancreatic cancer cells, including cell motility, proliferation, migration, neoangiogenesis, invasion, and apoptosis, presenting several quantity and quality aberrations, that can be exploited for the development of new biomarkers [167]. 

### 10.1. EVs in Pancreatic Juice 

It has been reported that EVs isolated from pancreatic juice (PJ) could be better utilized as diagnostic tools for PDAC patients, as PJ exosomal profiling could differentiate PDAC samples from other ones that are originated from benign and pre-malignant pancreatic diseases, with a high diagnostic accuracy (up to 91%). 

►**EVs with P-glycoprotein (MDR1), mucin (MUC) 1, MUC16, MUC5AC, and MUC6, as well as MUC4 and Cystic Fibrosis Transmembrane Conductance Regulator (CFTR)**, which are isolated from PJ, are considered diagnostic for PDAC [169]. ►**EVs with moesin, CD55, and Ras proteins** were found increased in the PJ of PDAC patients who were in the early stages (I–II), while patients in late stages (III-V) had an increased amount of EV-Ras [170]. ►**EVs with ADP-ribosylation factor 3, moesin, olfactomedin-4, pyruvate kinase, mucins, and Ras proteins, as well as CD55 and lipopolysaccharide-induced tumor necrosis factor:** They are members of a panel that includes 89 overexpressed proteins based on the study by Inoue H et al. (2022). These EVs are isolated from PJ that is collected via fine needle aspiration (FNA) samples guided by EUS. Their overexpression is considered diagnostic for PDAC among a population of patients with PDAC and Autoimmune pancreatitis (AIP) [171]. Meanwhile, in the aforementioned study, 64 EV-proteins were significantly reduced in PDAC patients, in comparison with the controls (AIP patients). ►**EVs with miR-155 and miR-21** were notably elevated in patients with PDAC, compared to patients with chronic pancreatitis (CP), as also demonstrated in the aforementioned study [171].

### 10.2. Circulating Blood EVs for Diagnosis and Screening


*Serum*
►**PLT-EVs (CD61 and CD41-positive) and CD63-positive EVs: Their levels were proved** diagnostic for PDAC [172]. It was proved that the levels of CD61, CD63, and CD41-positive EV in serum were higher in PDAC patients than in healthy controls, presenting an AUC of 0.846 [172]. Moreover, the performance of CA19-9 alone was compared to the EV (CD41+, CD63+, and CD61+) levels, as well as their combination, aiming the identification of the most diagnostic tool among them [172]. More particularly, the diagnostic accuracy of CA 19-9 alone, exhibited a lower one (AUC: 0.842) for PDAC discrimination from healthy patients, in comparison to serum CD61, CD63, and CD41+ levels [172]. However, their combination (CD63+ and CA19-9) showed a notably higher AUC of 0.903, which implies their potential role as an early-stage I–II diagnostic tool, compared to CA-19-9 alone (AUC: 0.814) [172]. Nevertheless, the above EVs had similar accuracy in the early (I–II) and late (III–IV) tumor stages. Furthermore, their post-operative levels were also studied, which were notably reduced in both cases, implying their proportional increase with the tumor growth and progression [172].►**EV with CD82+, GPC1, and levels of CA19-9:** This panel was studied by Xiao D et al. for its diagnostic accuracy in the early stages of PDAC in the Chinese population [173]. Favorably, the diagnostic accuracy of the aforementioned panel as a screening tool was significantly high (AUC = 0.942) [173]. Additionally, these researchers also studied the controversial role of EV-GPC1 alone as a screening biomarker [173].►**EVs with proto-oncogene mesenchymal–epithelial transition factor (c-met)** present a specificity of 85% and sensitivity of 70% [174].►**EVs A Disintegrin And Metalloproteinase (ADAM) 8 (ADAM8):** Its high levels are correlated with pre-malignant pancreatic lesion and PDAC [175]. ►**EV-ANXA6:** Its levels present a great AUC of 0.979 for detecting PDAC patients [176].►**EV-ZIP4**: Its high levels present a great diagnostic efficacy (AUC of 0.893) [119].►**EV-GPC1 and LRG-1:** This panel presents a great diagnostic performance, even for early-stage PDAC tumors with an AUC of 0.95 [177]. ►**EVs with RAS-associated protein RaB5 and D63 that are isolated by** ExoChip are considered potent diagnostic tools in serum samples [178]. ►**EVs with miR-21 and miR-17-5p:** EV-miR-21 has a AUC of 0.897 and EV-miR-17-5p has an AUC of 0.887 [179]. These EVs are found in high levels in PDAC patients. ►**EV-miR-1226-3p** has an AUC of 0.74 and it is downregulated in PDAC [180]. ►**EVs with miR-451a, miR-191, and miR-21** are found notably overexpressed in cases of IPMNs and PDAC, while they have an AUC of 0.759, 0.788, and 0.826, respectively [181].►**EVs with miR-4306, miR-1246, miR-3976, and miR-4644: This panel presented** an adequate specificity (AUC: 0.80), with their levels found increased in the majority of PDAC patients (83%) [182]. ►**EV-GPC-1 mRNA** have expressed levels in PDAC patients, independently of their tumor stage (AUC: 1.00) [183]. ►**EVs with CRNDE or MALAT-1 (lncRNAs)** were significantly increased in PDAC patients [176].►**EV-HULC (lncRNA):** Its levels were increased in IPMN or PDAC patients, in comparison with healthy controls, with great specificity (AUC: 0.92) [184]. 

Plasma►**EV-miR-10b**: Its level has been increased in PDAC patients, compared to healthy controls or CP patients (AUC: 0.81) [185].►**EVs with miR-205-5p, miR-122-5p, and miR-125b-3p:** This panel tested these EVs, which were found downregulated in Brazilians with PDAC. These EVs had an AUC of 0.857, 0.814, and 0.782, respectively [186].►**EVs with miR-451a, and miR-196a**: This panel has an AUC of 0.81, while the EVs have been tested as diagnostic biomarkers in several studies [187,188]. The former has been tested in early-stage PDAC (I–II) and has been proven that its level is statistically significant for the discrimination between stage I and II (*p*-value of 0.041) [187], while the latter, EV-miR-196a, has also been tested in patients at early disease stages (I–II stages) [188].►**EVs with miR-30c, EV-miR-10 b, miR-let7a, miR-21, and miR-181: This panel was** superior to the one of GPC1 [189]. Their levels have been significantly altered, including the low expression of EV-miR-let7a and the upregulation of the others, while it presented a great specificity in differentiating the healthy patients, or those with benign pancreatic disease (CP) from the ones with PDAC (AUC: 1.00) [189].►**EVs with miR-409 and mRNAs (CK18, CD63), combined by CA-19-9 and cell-free DNA concentration levels:** This multianalyte panel by Yang Z et al. (2020) has a great specificity, sensitivity, and accuracy of 95%, 88%, and 92%, respectively [190]. Meanwhile, plasma levels of EV-miR-409 constitute a potential diagnostic tool for PDAC with an AUC of 0.93. Additionally, the levels of EV-CK18 and EV-CD63 mRNAs were also increased in PDAC cases, with an AUC of 0.93 [190]. ►**EVs with long RNAs (TIMP1, FGA, HIST1H2BK, CLDN1, and ITIH2, as well as MAL2, MARCH 2, and KRT19):** The panel by Yu et al. (2019) that is based on long RNA sequencing presents a great AUC of 0.949 [191]. ►**EV-circ-IARS:** Its level have been overexpressed in PDAC tissues [192]. ►**EVs with EpCAM, GCP-1, and CD44V6:** This panel presents great specificity and an AUC of 1.00 [193]. ►**EVs with mutant proteins KRAS and/or P53** were detected in early-stage PDAC patients [194].►**EV-GPC1** alone did not have a high AUC (0.59) for PDAC detection [114].►**EVs with EphA2, EpCAM, and MIF:** This panel constitute great diagnostic tools for PDAC detection [195]. ►**EVs with WNT2, EpCAM, MUC1, GPC1, and EGFR:** This panel constitute great diagnostic tools for PDAC detection, presenting a specificity and a sensitivity of 81% and 86%, respectively [196].►**EV-alkaline** phosphatase **placental-like 2 (ALPPL2)** constitutes another diagnostic biomarker for PDAC, as was demonstrated in PDAC cell culture media [197]. 

Total Blood►**EVs with CD63 and GPC1** are increased in PDAC blood samples, with a high sensitivity of detecting PDAC (99%) and a specificity of 82% [198].

### 10.3. EVs in Saliva

►**EVs with miR-1246 and miR-4644** are significantly elevated in the saliva of PDAC patients (AUC 0.814 for miR-1246 and 0.763 for miR-4644), implying the potential use of this panel as a non-invasive diagnostic procedures [199]. ►**EVs mRNAs (Incenp, Apbb1ip, BCO31781, as well as Foxp1, Aspn, Daf2, and Gng2)** were significantly increased in PDAC mice models [200]. 

Last but not least, it has to be underlined the significance of exosomal miRNA profiles in PDAC patients, compared to CA-19-9 or the levels of EV-GPC1 for the differentiation of PDAC patients [201]. 

 ►**EVs with DNA that contain KRAS and TP53 mutations:** The PCR-mediated identification of EVs that contain several mutations such as KRAS and TP53, which are embedded with DNA molecules that present the aforementioned genetic aberrations, is a diagnostic method that could differentiate the health controls or CP patients from PDAC ones [202].

## 11. EVs as Prognostic Tools in PDAC 

In this section, we will present some of the prognostic EVs for PDAC patients [203].

Plasma and serum

The recent meta-analytic data in the study by Bunduc et al., which included 634 patients from eleven studies, demonstrated several exosomal biomarkers that were correlated with an elevated mortality risk, with their pre-operative levels being increased in resectable cases, compared to non-resectable ones [204]. 

►**Serum EpCAM-positive EVs, the total plasma EV concentration, and the levels of EV-DNA (KRAS)** did not increase the mortality non-resectable PDAC patients, implying that the detection of these biomarkers is not correlated with the resectability-based survival and mortality [204].►**Exosomes with phosphatidylethanolamine and miR-45** levels have been correlated to the resectability of PDAC [204].►**Exosomes with EpCAM, miR-200b, mir-222, and miR-451a:** This panel can be used in non-resectable cases for the selection of these patients, for whose systemic treatment plans could be advantageous [204].►**EV-Integrin α6:** Its high levels in PDAC patients have been closely associated with clinical recurrence, even months before the time of recurrence, whereas it has also been demonstrated that these levels were notably decreased postoperatively [205].

Plasma►**Plasma EV-Sox2ot (lncRNA)** levels have been significantly associated with PDAC progression, related to vascular and lymphatic dissemination of cancer cells [206].►**Plasma EV-Circ-PDE8A** levels were significantly overexpressed in PDAC [207].►**Plasma EV-circ-IARS** overexpressed levels have been detected in advanced (metastatic) PDAC cases (*p*-value of 0.002) [208].►**Plasma EV-miR-222** has been associated with tumor invasion and metastasis in PDAC culture media, as well as the survival of PDAC cancer cells by down-regulating p27 and suppressing PPP2R2A expression, with the latter leading to AKT pathway activation [95]. The aforementioned EV-miRNA could be utilized as a prognostic factor for survival, tumor stage, and size [95].►**Plasma EVs contacting MIF, GPC1, EpCAM, and CD44V6** [204].►**Plasma EV-MIF** levels are found increased in PDAC patients that do not present liver metastasis [209].►**Plasma EV-GPC-1** levels are correlated to tumor burden and size [201].►**Plasma EVs with EpCAM and CD44V6** also have a prognostic role in PDAC, with the former’s levels being associated with the treatment response during systemic, palliative chemotherapy in advanced PDAC cases, as well as with the tumor stage [210], whereas the latter’s are only related to the tumor stage [193].

Serum►**Serum EVs with C1QBP and CD44V6** were found increased in PDAC patients, and they are associated with prognosis and liver metastatic disease [211].

In Table 5, we present a summary of EVs that are considered diagnostic and/or prognostic tools in PDAC.

## 12. EVs as Therapeutic Tools in PDAC

Despite the great progress in the PDAC therapeutic strategies, PDAC remains a chemoresistant tumor. There is a need of developing novel agents, new druggable targets, as well as innovative means of drug delivery, including EVs. These nanocarriers have a low antigenicity, a strong specificity and they can deliver chemotherapeutic agents or RNA molecules in the targeted sites, overcoming the intense desmoplasia that PDAC tumors present. There are has been significant progress in the field of EV-based therapy in PDAC cases [167], while several EV-based monitoring tools for treatment response have also been demonstrated. More particularly, IgG-positive EVs are reduced when there is a favorable response to treatment, whereas they are increased in case of tumor progression. This marker is considered beneficial for the identification of treatment responses in patients who do not express high levels of CA-19-9 [212].

### 12.1. EVs as Vectors for Bioactive Molecules in PDAC Treatment

EVs can carry not only drugs but also a high variety of molecules that can target molecular pathways, which induce tumorigenesis, or that can activate tumor inhibitory genes, as well as suppress oncogenes [213]. An example of the aforementioned application of EVs is the EV-assisted targeting of mutant GTPase KRAS. It has been reported that engineered EVs (iExosomes) from physiological fibroblast-like mesenchymal cells carry as cargo, a short hairpin or interfering RNA molecule that targets the oncogenic KRAS. It has to be underlined that EVs have a superior efficacy compared to liposomes, as was demonstrated in xenografts (mice) [213].

Another similar application of iExosomes is the one that includes the utilization of EVs that derive from mesenchymal stromal cells of the human umbilical cord (UC-MSC-derived EVs). These EVs are embedded with siRNA or therapeutic agents that target the PDAC cells [214]. More particularly, they are loaded with KRASG12D-targeting siRNA, resulting in a notable decrease in the expression of KRASG12D. The uptake of KRASG12D siRNA EVs by the cell lines such as LS180, BxPC-3, and PANC-1 induced the apoptosis of the lines that were expressing KRASG12D (LS180 and PANC-1) [214].

Furthermore, EVs can also facilitate and enhance the effect of immunotherapy, which has a promising role in PDAC treatment management, as it significantly improves the survival of the patients [215]. Enhancement of the anti-cancer immune response is the key for the optimal PDAC management, which can be mediated via the utilization of bone marrow mesenchymal stem cell (BM-MSC) exosomes that can be loaded with galectin-9 siRNA via electroporation, as well as with prodrug oxaliplatin (OXA), the so-called iEXO-OCA [215]. The administration of these iExosomes can inhibit the immunosuppressive effect of TAMs by interrupting the axis of galectin-9/dectin, while via OXA, it is possible to deregulate Tregs and recruit T-cytotoxic cells in the PDAC microenvironment. The aforementioned phenomenon implies the beneficial effect of iExosomes in PDAC-TME reprogramming [215].

### 12.2. EVs as Drug Vectors in PDAC Treatment

Exosomes constitute a forward-looking plan for drug delivery, as they can transport several cargoes, which are endocytosed by the neighboring PDAC cells.

In the study of the engineered UC-MSC-derived EVs that were previously mentioned, the researchers loaded these EVs with doxorubicin (DOXO), which constitutes a chemotherapeutic agent. The recipient cells of these EVs were several cell lines, including LS180 that express KRASG12D (colorectal cell line), BxPC-3, and PANC-1 (the former expressing KRASwt and the latter, KRASG12D). The uptake of these DOXO-loaded EVs by PANC-1 cells, induced their death, compared to free administrated DOXO [214].

Moreover, the therapy that utilizes GEM-loaded autologous exosomes, the so-called ExoGEM therapy, has been demonstrated as a potential chemotherapeutic strategy for PDAC patients. More particularly, it is reported that this type of chemotherapy is safer and more effective than the systemic type in animal PDAC models (mice) [216]. Additionally, it has been demonstrated that PDAC growth was significantly inhibited after the administration of ExoGEM, and the survival of the mice was notably elongated, with both of these results being dose-related. In addition, it has to be noted that some other advantages of ExoGEM are the lack of PDAC recurrence, its higher efficacy (more targeted), and its low toxicity [216].

Furthermore, it has also been demonstrated that the loading of exosomes with Pirfenidone, which is an agent that is widely used in idiopathic pulmonary fibrosis (IPF), can act as an anti-fibrotic agent in pre-metastatic niches. Taking advantage of the effect of EV in the activation of the fibrotic pre-metastatic niches, the delivery of Pirfenidone-loaded exosomes that are secreted by PDAC cells, could significantly suppresses liver metastasis, implying its potential use as a strategy against tumor metastatic dissemination [217].

### 12.3. EVs as Targets

The targeting of cancer-related EVs, including PDAC-EVs is considered a major challenge. For example, the utilization of the tetraacetylated N-azidoacetyl-d-mannosamine-loaded nanoparticles could open new therapeutic chances, with the targeting and tagging of PDAC cells and their secreted EVs [218].

Additionally, a bio-orthogonal click reaction induces the suppression of PDAC cells and EVs via the utilization of another type of modified nanoparticle, namely dibenzyl-cyclootyne. The aforementioned strategy plan inhibits the proliferation of PDAC cells, but also the oncogenic role of their secreted EVs [218].

Another therapeutic strategy that has been demonstrated not only in vivo but also in vitro is the suppression of EV-VEGF-C expression by using the selective HDAC1/2 inhibitor (B390), which also inhibits the proliferation of endothelial cells in lymphatic vessels (it inhibits lymphangiogenesis and lymphatic tumor dissemination) [219]. Moreover, B390 restores the altered functionality of dual specificity phosphatase-2 (DUSP2), which constitutes an ERK phosphatase that is implicated in ERK signaling (commonly overactivated in PDAC tumors due to mutant KRAS) [219]. Additionally, the exosome inhibitor GW4869 targets EV-mir-21, which deregulate PTEN expression, leading to chemoresistance and PDAC progression. In addition, GW4869 restores PTEN expression in vivo and in vitro [220]. Likewise, inhibiting the CD9-positive ANXA6-EVs could potentially reduce the stromal modification and PDAC tumor progression [140].

### 12.4. EV-Based Photodynamic Therapy and EV-Based Immunotherapy

Τumor-derived re-assembled exosomes (R-Exo) can be utilized in phototherapy as a carrier of photosensitizer, such as clorin e6, the so-called PDAC-derived CeR-R exo. These vesicles are not only increasing the photosensitivity of the tumor cells but also have a stimulatory effect on immune cells, via inducing cytokine release by immune cells. Moreover, these Ce6-R-Exo induce an increase in reactive oxygen species in PDAC cells, under the effect of the irradiation laser, a phenomenon that can be visualized via the so-called photoacoustic imaging [221]. In Table 6, we summarize some of the EV-based treatment strategies for PDAC management.

## 13. Pros and Cons of EV-Utilization in Research

Several ongoing studies are demonstrating the new diagnostic and therapeutic challenges of EV engineering. However, several limitations are considered obstacles to EV utilization in scientific research, and there is a need to overcome them. Firstly, a −80 ◦C storage environment is usually required for their preservation, which permits the enhancement of their stability and functionality. The aforementioned method of storage, so-called cryopreservation, gives rise to many obstacles, including the high expenses required for their shipping, the expensive storage equipment, as well as the many modifications needed to maintain the function and integrity of EVs due to the freeze–thaw cycles [222]. Nevertheless, the side effects of cryopreservation can be resolved via the utilization of alternative storage methods such as spray-drying, lyophilization, or freeze-drying procedures. The former technique requires the use of an antifreeze substance, so-called trehalose, that limits the aggregation effect of the EVs during their freeze–thawing procedures, whereas the latter method permits the stabilization of the vesicles via their conversion into a dry structure, so-called powdered exosomes. Another major limitation in the utilization of EVs is the lack of any standardized protocol for their isolation, with each method presenting several pros and cons. One of the most used isolation method is ultracentrifugation, for which the main con is the possible breakage of EVs due to serial centrifugations [222,223]. Despite all the aforementioned limitations, EVs remain an optimal tool for nanoscale-based delivery methods in cancer treatment, such as drug-embedded exosomes. This is especially due to their low antigenicity and high biocompatibility.

There are several EV-based strategies for anti-cancer treatment such as the delivery of EVs and their cargoes (chemotherapeutic agents, nucleic acids, or protein molecules), which have an anti-tumor effect [224]. It has to be underlined that the mass production of engineered EVs is quite unfeasible. However, this issue can be solved via the utilization of exosome-mimetic or EVs-liposome hybrid nanovesicles, or even via the use of natural product-derived vesicles that arise from plants such as fruits or milk. More specifically, it is reported that EVs are considered superior to liposomes as drug delivery systems, which is mainly attributed to their rapidly increased expression levels in the circulation, as well as their low toxicity [225].

Meanwhile, their pivotal role in intercellular communication through their cargoes, as well as their presence in several biological fluids, makes them attractive tools for the non-invasive diagnosis of several diseases including cancer [226]. Despite the various pros of utilization as diagnostic biomarkers in different biological materials, as they constitute nanocarriers of several molecules for intercellular communication, they also have some significant cons. First of all, an adequate amount of EVs may require a big sample of the biological material, due to their possible destruction by serial centrifugations for their isolation.

Finally, the aforementioned limitations could be solved via the standardization of the circulating EV levels for PDAC-related management and therapeutic decisions, as well as via the development of clear guidelines for their isolation and storage [227]. Further research about their utilization needs to be undertaken, and clinical trials need to be conducted.

## 14. Future Perspectives of EV-Based Machine Learning-Based Algorithms

Machine learning (ML)-based algorithms are part of the major field of Artificial Intelligence (AI) and have broadened the horizons of PDAC detection via the integrative analysis of several components of PDAC diagnosis, including EV-based diagnostic biomarkers and clinical datasets, as well as digitized optical imaging and multi-omics [228,229]. The utilization of the aforementioned strategy via AI and machine learning-based algorithms will potentially facilitate PDAC detection in asymptomatic populations or in early stages, when the disease can be surgically treated. An emerging field of EV studies is EV microscale cytometry using tissue and distinct biomarkers for the disease, with the goal being to perform a complex analysis and extract predictive models. The aforementioned ML analysis is the so-called extracellular vesicle machine learning analysis platform (EVMAP), which constitutes a useful tool for prediction, by using blood samples [230].

## 15. Conclusions

EVs, as well as non-coding RNAs, have a quite promising role in the development of diagnostic, and prognostic tools. Especially with the contribution of multi-omics, machine learning algorithms, and AI, providing an integrative analysis that facilitates the PDAC diagnosis in early or asymptomatic stages. Meanwhile, their crucial role in intercellular communication through their cargoes opens up new therapeutic approaches via their utilization as nanocarriers of anti-neoplastic therapeutic agents or bioactive molecules that induce tumor suppression opens up new therapeutic perspectives for this highly aggressive GI tumor. However, the significant cons of their utilization should be overpassed via the standardization of their isolation procedures and the limitation of their damaging through freeze–thawing.

## Figures and Tables

**Figure 1 ijms-25-03406-f001:**
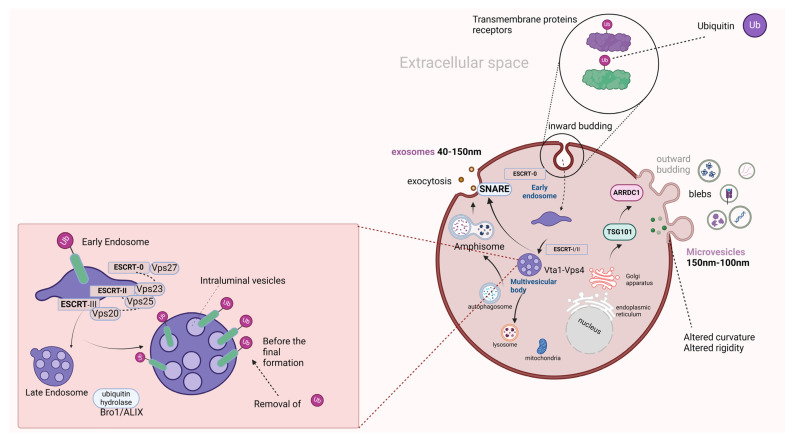
A schematic presentation of EV biogenesis routes and the ESCRT-dependent exosome biogenesis. Exosomes are either produced via an ESCRT-dependent or independent pathway (cargo trafficking from cytosol or trans-Golgi complex). In this scheme, we present the dependent one in detail, starting with inward membrane budding and the internalization of the ubiquitinated transmembrane proteins or receptors, leading to the early endosome formation, which further matures into the late endosome. ESCRT-0 is required for the identification of these proteins, the binding on the endosomal membrane, and the recruitment of ESCRT-I. The latter recruits further ubiquitinated proteins and recruits ESCRT-II. Later, ESCRT-III is recruited by ESCRT-II and contributes to vesicle splitting from MVB. SNARE proteins are required for MVB fusion on the membrane for the release of exosomes. Microvesicle biogenesis requires the modification of the cell membrane (blebbing) via the interaction between ARRD1 and TSG101. These microvesicles could contain DNA, RNA, protein molecules, and receptors, as it is demonstrated. This figure was created with “BioRender.com”, accessed on 11 March 2024 (Agreement number CW26K9TIYO).

**Figure 2 ijms-25-03406-f002:**
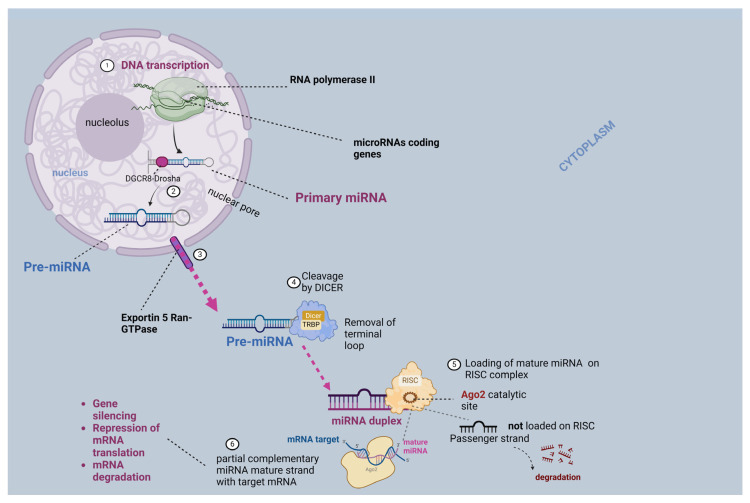
A schematic presentation of the procedure of miRNA biogenesis. This pathway starts with pri-miRNA formation via the transcription of specific miRNA coding genes, which is further cleaved by DGCR8–Drosha complex, leading to precursor miRNAs (pre-miRNA). The latter are exported from nuclear pores via Exportin 5 and Ran-GTP complex, towards the cytoplasm, and then cleaved by the Dicer–TRBP complex, enabling the loading of miRNA duplexes on RISC. The presence of a binding site in duplex-miRNA for the catalytic site on RISC leads to the separation of the two strands into the mature (active strand) and the passenger, with the latter being degraded. MiRISC binds on the 3′untranslated region of the targeted mRNA strand that is partially complementary, leading to gene silencing, including the repression of mRNA translational procedure, or its degradation. This figure was created with BioRender.com (accessed on 11 March 2024 agreement number ZD26KA094M).

**Table 1 ijms-25-03406-t001:** OncomiRs in PDAC.

miRNA	Role in PDAC	Expression Levels
miR-21 [36,37]	↑ proliferation via targeting Ras-Raf-MEK-ERK PI3K/AKT EGFR signaling pathways ↓ apoptosis	Increased
miR-186 [38]	↑ proliferation via targeting NR5A2	Increased
miR-17-5p [39]	Decontrolled cell cycle ↑ proliferation via targeting RBL2/E2F4 repressing complexes	Increased
miR-196b [40]	↑ proliferation Apoptosis suppression via targeting CADM1	Increased
mir-18a [41]	Higher expression in pancreatic cancer tissues Lower in postoperative samples Member of miR-17–92 cluster ↑ genome proliferation Transcriptional activation by MYC	Increased
miR-191 [42]	Extracellular matrix modification ↑ metastasis	Increased
miR-29a [43,44]	Migration and invasion	Increased
miR-221 [45]	↑ proliferation and ↓ apoptosis Metastatic dissemination	Increased
mir-301a-3p [46]	↑ migration and invasion via targeting SMAD4	Increased
miR-374a [47]	↑ migration and proliferation, EMT via ↓ SRCIN1	Increased
miR-1469-5p [48]	↑ proliferation and progression via targeting NDRG1/NF-κB/E-cadherin axis	Increased
mir-205 [49]	↑ proliferation Targets *APC* Implicated in Wnt/β-catenin signaling pathway	Increased
miR-10b [50]	↑ invasive behavior, progression TIP30 expression and ↑ EGF and TGF-β effects	Increased

Secernin 1 (SRCIN1), tumor growth Factor β (TGF-β), epidermal growth factor (EGF), epithelial–mesenchymal transition (EMT), Tat-interacting protein 30 (TIP30), ↑ increase, ↓ decrease.

**Table 2 ijms-25-03406-t002:** Tumor-suppressive miRNAs in PDAC.

Tumor-Suppressive miRNAs	Role in PDAC	Expression Levels
miR-506 [51]	↓ tumor growth and progression	decreased
miR-34 [52]	inhibition of the cancer stem cells	decreased
miR-142 [53]	↓ tumor growth and invasion via regulation of HIF-1a expression	decreased
miR-216b [54]	Regulation of TC1TP expression KRAS inhibition	decreased
miR-30c [55]	↓ tumor growth and progression G1 phase and apoptosis regulation	decreased
miR-143-3p [56]	↓ malignant transformation of pancreatic cells MERK/ERK signaling suppression	decreased
miR-519-3d [57]	↓ hypoxia-induced tumorigenesis by regulating ICI, PD-L1	decreased
miR-1181 [58]	↓ tumor invasion and progression	decreased
miR-375 [59]	↑ apoptosis, ↓ lymphatic spread ↓ metastasis	decreased
miR-455-3p [60]	↓ growth, ↑ apoptosis, EMT, regulation of TAZ expression	decreased
miR-135a [61]	Growth and progression bmi1 targeting	decreased
miR-340 [62]	↓ tumor growth and progression, regulation of BICD2 expression	decreased
miR-203a-3p [63]	↓ growth, invasion, limitation of EMT, regulation of FGF2 expression	decreased

Hypoxia-inducible factor 1-alpha (HIF-1a), epithelial–mesenchymal transition (EMT), translationally controlled 1 tumor proteins (TCPT1), Ribosomal Protein S15a (RPS15A), Bicaudal-D2 (BICD2), fibroblast growth factor 2 (FGF2), ↑ increase, ↓ decrease.

**Table 3 ijms-25-03406-t003:** PDAC-derived EV-miRNAs and EV-proteins with their distinct effect and role in PDAC progression.

Parental Cell	Cargo	Effect
PDAC	miR-197-3p miR-6796-3p miR-4750-3p miR-6763-5p	Deregulation of glucose metabolism via GIP and GLP1 altered expression [91]
	miR-125b-5p miR-450b-3p miR-666-3p miR-883b-5p	Interrupting PI3K/Akt/FoxO1 axis Insulin resistance in C2C12 Myotube cells [91]
	miR-540-3p	
	hsa-miR-3133 hsa-miR-144-5p hsa-miR-3148	Insulin resistance and/or DM-associated PDAC [93]
	miR-19a	Altered insulin production, via interacting with the Neurod1 protein implicated in DM-associated PDAC [94]
	miR-222	Promotes cancer proliferation, which is correlated with the size and stage of tumor [95]
	miR-155	Suppression of apoptosis GEM chemoresistance Promotion of cancer cell proliferation [96]
	miR-125b-5p	Activates the MEK/ERK pathway, promotes PDAC invasion, EMT, metastatic dissemination [97].
	miR-27a	↑ angiogenesis through BTG2 PDAC invasion, human microvascular endothelial cells (HMVEC) angiogenesis progression and metastasis [98]
	CKAP4 O-glycan-binding lectin	↑ pre-operative levels ↓ post-operative [100,101]
	B2M	Tumor escape phenomenon [102]
	EGFR KRAS CD44	PDAC progression Shorter overall and progression-free survival (↑ EV-KRAS) [102]
	CAV1 CLU	Over-proliferation Impaired apoptosis of pancreatic cancer cells [102]
	ITGA3 PODX	PDAC progression, migration, invasion of surrounding tissue [102,103,104,105,106]
	Tspan8 Integrins CD151	Stromal changes, promoting cell motility, invasion and metastatic capacity [107]
	ITGΒ5 S100A4 ANXA1 F3	Facilitate the metastatic dissemination [102,109] (pre-metastatic niche)
	STAT14 LAMP1 Lin28B	[102]
	MIF GFP + EVs	↑ MIF in early stages of PDAC progression and ↑ cytokines in patients at stage I, who will develop liver metastasis [110,111]. Distant metastasis in mice Increased vascular permeability and induction of HUVEC cells in vitro [112].
	GPC1	Poor prognosis and decreased survival [113,114].
	EphA2	↑ proportional to the tumor stage Prediction for neoadjuvant treatment response and possible monitoring tool [115]
	MUC1 CLDN1	PDAC progression and poor prognosis [102,116,117]
	HIST2H2BE CD151 CLDN4 LGALS3BP EpCAM	Overall, 73% of exosomes with the selected markers had KRAS mutation [118]
	ZIP4	Oncogenic potential for non-malignant cells found upregulated in highly malignant PDAC [119]
	Adrenomedullin	Marker for β-cell destruction in PDAC [120]

↑ increase, ↓ decrease, cytoskeleton-associated protein 4 (CKAP4), β2-microglobulin (B2M), epidermal growth factor receptor (EGFR), Caveolin-1 (CAV1), Clusterin (CLU), integrin (ITGA3), Podocalyxin-like protein (PODX), Integrin Beta-5 (ITGΒ5), Annexin A1 (ANXA1), tissue factor 3 (F3), migration inhibitory factor (MIF), Glypican-1 (GPC1), Mucin 1 (MUC1), claudin-1 (CLDN1).

**Table 4 ijms-25-03406-t004:** The implication of EVs in PDAC-TME based on their origin, cargo, and recipient cells.

EV-Origin	Cargo	Recipient Cells	Effect on PDAC TME
PDAC	EVs	T cells	T cell death↑ via p38 MAPK pathway and the initiation of Endoplasmic reticulum stress [124]
	Integrins	Nk cells	↓ functionality of NK cells via ↓ expression of INF-γ, CD107a, and TNF-α in NK cells [125]
	A3 adenosine receptors	Mast cells	Induction of MAP and ERK1/2 kinases, PDAC progression [126]
	ANXA1	ESCs fibroblasts	Angiogenesis, activation of FPRs [109] Implicated in the Akt/ERK pathway, neoangiogenesis [127]
	MiR-27a	ESCs	BTG2 expression—increased proliferation and angiogenesis [98]
	hnENPA1		Lymphangiogenesis and lymphatic metastasis [98]
	EVs	non-malignant pancreatic cells	UPR and ER stress in 24 h via targeting the expression of DDIT3 [129]
	palmitic acid		ER stress in physiological pancreatic cells [129]
	EZR	TAMs	Altered macrophage polarization and PDAC progression [130]
	miR-155-5p		Interaction with EHF/Akt/NF-kB pathway—altered M2 phenotype [131]
	FGD5-AS1.		Activation of the STAT3/NF-κB pathway, worsening the prognosis in PDAC patients [133]
	Lin28B	PSCs	Recruitment of PSCs via overexpression of PDGFB, promotion of metastasis [132]
	miR-125b-5p	stromal	MEK/ERK pathway induces PDAC invasion, EMT, as well as metastatic dissemination [97]
			Suppression of STARD13 targeting—deregulation of regulation of interorganelle cholesterol transportation PDAC progression
	LDL receptor-related proteins		Activation of YAP PDAC progression [136]
CAFs	miR-21	PDAC cells	Chemoresistance and PDAC progression PTEN inhibited expression [137,138]
	miR-3173-5p	PDAC cells	GEM resistance in xenograft PDAC mouse model, via ACSL4 gene Suppression of ferroptosis in PDAC cells [139]
	ANXA6		PDAC aggressive behavior via the overexpression of CD9 on the EV surface Induction of MAPK pathway activity, increases EMT, and promotes PDAC expansion [140]
	MiR-106b		GEM resistance via its interaction with TP53INP1 [141,142]
Blood	EVs		↑ levels, independently interrelated with increased progression-free survival and PDAC control rate for patients under chemotherapy [144]
WBC	EVs		↑ levels in unresectable or borderline resectable PDAC patients—an independent factor for increased overall survival [144]
PSCs	miR-451a miR-21-5p		Promotion of the PDAC progression and expansion via promoting the expression of CCL1, and CCL2 [145]
TAMs	miR-21a-5p miR-501-3p miR-301a-3p		PDAC progression [90] Interaction with TGFBR3 via the activation of the TGF-β pathway [146,147], M2 phenotypic state promotes PDAC progression to metastasis via the PTEN/PI3Kγ pathway [148]
PDAC	TF		Promotion of thrombosis in PDAC patients [154]
	circPDK1		HIF1A induces their secretion, PDAC cell survival, and migration via c-myc stimulation, promoting glycolysis in vivo [158]
BxR-CSCs	miR-210	PANC-1/BxS	Resistance to GEM [160]
GIPC-depleted PANC1 or AsPC-1	ABCG2	PDAC	Resistance to GEM [161]
PANC1	EphA2	BxPC-3/MiaPaCa2	Resistance to GEM [115,162]
PANC1, MiaPaCa2	miR-155		Resistance to GEM [96,162]
BxPC-3-Gem cells	MMP14	BxPC-3/MiaPaCa2	Resistance to GEM [162]
CAF	Snail-Mrna miR-106b	ECs	resistance to GEM [138,162]
PDAC GEM-treated	miR-155 CAT SOD2		resistance to GEM in vitro [95,168]
PDAC	miR-155	PDAC cells	Implicated in GEM metabolism and inactivation, via suppressing the GEM-metabolizing enzyme DCK in PDAC cells [165]
	ZIP4		Leading to cachexia [166], induce PDAC progression [119]

↑ increased; ↓ decrease; PDAC; pancreatic ductal adenocarcinoma; CAFs, cancer-associated fibroblasts; TAM, tumor-associated macrophages; gemcitabine, GEM; EphA2, Ephrin type-A receptor 2; ABCG2, the ATP-binding cassette (ABC) superfamily G member 2; TP53INP1, tumor protein 53-induced nuclear protein 1; MMP14, matrix metalloproteinase 14.; CSCs, cancer stem cells; PSCs, pancreatic stellate cells; WBC, white blood cells; ECs, endothelial cells; UPR, unfolded protein response1; ER, endoplasmic reticulum.

**Table 5 ijms-25-03406-t005:** EVs as diagnostic and/or prognostic tools in PDAC.

Role	Sample	Cargo Family	Cargo	Characteristics
Diagnosis	PJ	Proteins	MDR1, MUC 1, MUC16, MUC5AC, MUC6, MUC4, CFTR	Differentiation of PDAC from other benign and premalignant pancreatic diseases (diagnostic accuracy of up to 91%) [169]
	PJ	Proteins	Overregulated ARF3, MSN, OLFM4, PK, MUC, Ras, LITAF	Panel from EV-proteins from PJ collected by EUS-FNA. In the early PDAC stages (I–II), MSN, CD55, and Ras proteins were notably overexpressed, with Ras being overexpression in PDAC stages III and IV compared to AIP patients and HC [170]
	PJ	MiRNA	miR-155 miR-21	Notably elevated in patients with PDAC, compared to patients with CP [171]
Diagnosis	Serum	Proteins	CD61, CD41, CD63	↑ levels of PLT-EVs, CD61+, CD41+, and CD63+ EVs are higher in PDAC patients than in HC (AUC 0.846) Combination with CA-19-9 (AUC 0.903 versus CA19-9 alone AUC: 0.814) [172]
			CD82+, GPC1	Combined with CA19-9 for early diagnosis (AUC: 0.942) [173]
		Proteins	c-met	↑ EV-c-met: specificity of 85% and sensitivity of 70% for PDAC diagnosis [174]
			ADAM8	↑ EV-ADAM8 correlated with pre-malignant pancreatic lesions and PDAC vs. HC [175]
			ANXA6	↑ EV-ANXA6 has AUC of 0.979 for detecting PDAC [176]
			ZIP4	↑ EV-ZIP4 has great diagnostic efficacy (AUC: 0.893) [176]
			GPC1, LRG-1	Presents a great diagnostic performance, even for early-stage PDAC tumors with an AUC of 0.95 [177]
			Rab5, D63.	Detected by ExoChip, as potent PDAC diagnostic tools [178]
	Serum	miRNA	miR-21 miR-17-5p miR-1226-3p miR-451a miR-191 miR-21	↑ EV-miR-21 for PDAC detection with AUC: 0.897 ↑ EV-miR-17-5p for PDAC detection (AUC: 0.887) [179] ↓ EV-miR-1226-3p for PDAC detection with AUC: 0.74 [180]. ↑ EV-miR-451a (AUC: 0.759), ↑ EV-miR-191(AUC: 0.788) and ↑ EV-miR-21 (AUC: 0.826) for PDAC and IPMN detection [181]
			miR-4306, miR-1246, miR-3976, miR-4644	Increased level of the panel EV-miRNAs in the majority of PDAC patients (83%) with an AUC: 0.80 [182]
		mRNA	GPC-1 mRNA	↑ EVs-GPC-1 mRNA in PDAC patients, independently of the stage (AUC: 1.00) [183]
		lncRNA	CRNDE MALAT-1 HULC	↑ EV-lncRNA with CRNDE or MALAT-1 were significantly increased in PDAC patients [176] ↑ EV-lncRNA HULC in IPMN or PDAC patients, in comparison with healthy controls, with great specificity (AUC: 0.92) [184]
Diagnosis	Plasma	miRNA	miR-205-5p miR-122-5p miR-125b-3p miR-10b miR-451a, miR-196a miR-451a	↑ EV-miR-205-5 p for PDAC detection with AUC of 0.857 ↑ EV-miR-122-5p for PDAC detection with AUC of 0.814 ↑ EV-miR-125b-3p for PDAC detection with AUC of 0.782 ↑ EV-miR-10b for PDAC detection with AUC of 0.81; former has been increased in PDAC patients, compared to healthy controls or CP patients [186] Increased levels for PDAC detection (AUC: 0.81) ↑ EV-miR-451a n PDAC patients, compared to HC or CP patients in early-stage PDAC (I–II) and has proven statistically significant for the discrimination between stage I and II (*p*-value of 0.041) [187,188]
Diagnosis	Plasma	miRNA	miR-196a	↑ EV-miR-196a in early-stage PDAC patients (I–II stages) [188]
			miR-30c, miR-10 b, miR-let7a, miR-21, miR-181	Up-regulated, except EV-miR-let7a, differentiating the benign pancreatic disease (CP) and healthy donors from PDAC patients (AUC: 1.00) [189]
	Plasma	miRNAs, mRNAs	CK18 mRNA CD63 mRNA EV-miR.409	Multianalyte panel consisting of CA-19-9, cell-free DNA, EV-miRNAs, and EV-mRNAs has a great specificity, sensitivity, and accuracy of 95%, 88%, and 92%, respectively [190].
		miRNA	has-miR-409	Diagnostic tool for PDAC with an AUC of 0.93 [190]
		mRNAs	CK18 and CD63 mRNAs	Increased in PDAC cases with an AUC of 0.93 [190]
	Plasma	Long RNAs	TIMP1, FGA, HIST1H2BK, CLDN1, ITIH2, MAL2, MARCH 2, KRT19	Presented in PDAC with AUC of 0.949 [191]
		circRNA	IARS	Overexpressed in PDAC tissues [192]
		Protein	EpCAM, GCP-1, CD44V6	Panel for PDAC detection with AUC of 1.00 [193]
		Mutant protein	Mutant KRAS and/or P53	Mutant proteins KRAS and/or P53 in EVs were detected in early-stage PDAC patients [194]
	Plasma	Protein	GPC1	EV-GPC1 alone do not have high AUC (0.59) for PDAC detection [114]
	Plasma		EphA2 EpCAM MIF WNT2, EpCAM, MUC1, GPC1 and EGFR	Detection of PDAC from HC [195] Panel presenting a specificity and sensitivity of 81% and 86%, respectively [196]
			ALPPL2	Diagnostic biomarker for PDAC, as it was demonstrated in PDAC cell culture media [197]
	Blood	Protein	CD63, GPC1	↑ EV-CD63 and ↑ EV-GPC1 levels in PDAC blood samples, with a high sensitivity of detecting PDAC (99%) and a specificity of 82% [198]
	Saliva	miRNA	miR-1246 miR-4644	↑ EV-miR-1246 levels in PDAC patients with AUC of 0.814 ↑ EV-miR-4644 levels in PDAC patients (AUC of 0.763) [199]
	Saliva	mRNA	Incenp, Apbb1ip, BCO31781, Foxp1, Aspn, Daf2, Gng2	Significantly increased in PDAC mice models [200]
Prognosis	Serum and plasma	Protein and DNA Lipid and miRNA	EpCAM (serum) KRAS- DNA (plasma) phosphatidylethanolamine miR-45	The levels of these biomarkers did not increase the mortality in non-resectable PDAC patients The levels of these biomarkers were correlated with increased mortality risk in resectable PDAC cases, facilitating the stratification of these patients, in which systemic treatment is advantageous [204]
		Protein and miRNA	EpCAM, miR-200b, mir-222, miR-451a	↑ pre-operative expression levels t are correlated with ↓ survival [204]
			KRAS	KRAS MAF ≥ 1% level during chemotherapy is correlated with PDAC progression before the increase in CA19-9 or the presence of radiological findings (≈50 days before) [204]
	Blood	protein	Integrin α6	Increased levels in clinical recurrence (even months before the time of recurrence) and decreased post-operatively [205]
Prognosis	Plasma	lncRNA	Sox2ot	↑ EV-lncRNA Sox2ot in PDAC progression, vascular, and lymphatic dissemination [206]
		circRNA	PDE8A	Overexpressed in PDAC [207]
	Plasma	circRNA	IARS	Increased in advanced (metastatic) PDAC cases (*p*-value of 0.002) [208]
		miRNA	miR-222	EV-miR-222 from PDAC patients promote tumor invasion and metastatic dissemination, and ↓ survival in PDAC cell cultures [95]
		protein	MIF GPC1, EpCAM EV-CD44V6	↑ EV-MIF are correlated to PDAC patients with no liver metastasis [209] ↑ EV-GPC-1 levels are correlated to tumor burden/size [201] ↑ EV-EpCAM levels are associated with the treatment response during systemic, palliative chemotherapy in advanced PDAC stage [210] ↑ EV-CD44V6 are associated with tumor size [193]
	Plasma	Protein	EpCAM, GCP-1, CD44V6	Plasma panel for prognosis (tumor stage) and diagnosis (AUC: 1.00) [193]
	Serum		C1QBP, CD44V6	↑ EV-C1QBP and EV-CD44V6 have been closely associated with prognosis in PDAC patients and for liver metastatic disease [211]

ARF3, ADP-ribosylation factor; MSN, moesin; OLFM4, olfactomedin-4; PK, pyruvate kinase; MUC, mucins; LITAF, lipopolysaccharide-induced tumor necrosis factor; AIP, autoimmune pancreatitis; CP, chronic pancreatitis; HC, healthy controls; ↑ increased; ↓ decrease; PDAC, pancreatic ductal adenocarcinoma; GEM, gemcitabine.

**Table 6 ijms-25-03406-t006:** EV-based treatment strategies for PDAC management.

EV-Based Modality	Cargoes	Therapeutic Target	Characteristics
Vector for bioactive molecules	Short hairpin or interfering RNA molecule	Mutant GTPase KRAS,	Engineered EVs from physiological fibroblast-like loaded with a short hairpin or interfering RNA molecule for KRAS mutation. PDAC suppression was reported in xenografts (mice) [213].
	SiRNA (KRASG12D-targeting siRNA) or therapeutic agents (DOXO)	PDAC cells with KRASG12D expression	Mesenchymal stromal cells of the human umbilical cord (UC-MSC-derived EVs) that target the PDAC cells, resulting in a notable decrease in the expression of KRASG12D. The uptake of KRASG12D siRNA EVs by the cell lines induced the apoptosis of the lines that were expressing KRASG12D (the LS180 and PANC-1). Improvement of survival and enhancement of the anti-cancer immune response [214].
	galectin-9 siRNA and OXA	TAMs	Bone marrow mesenchymal stem cell (BM-MSC) exosomes, which can be loaded with galectin-9 siRNA and prodrug oxaliplatin (OXA). Inhibition of the immunosuppressive effect of TAMs interrupting the axis of galectin-9/dectin [215]. OXA–related deregulation of Tregs and recruitment T-cytotoxic cells in the TME (reprogramming) [215].
Drug vector	GEM	PDAC cells	GEM-loaded autologous exosomes. PDAC growth was significantly inhibited. Mice survival was elongated, with dose-related results, and more targeted and less toxic treatment [216].
	Pirfenidone		Anti-fibrotic agent in pre-metastatic niches. Delivery of Pirfenidone-loaded exosomes derived from PDAC cells. Significantly suppresses liver metastasis [217].
EV as target	Agents for EV targeting: tetraacetylated N-azidoacetyl-d-mannosamine-loaded nanoparticles and modified nanoparticles (dibenzyl-cyclootyne). Selective HDAC1/2 inhibitor (B390)	PDAC cells EV-VEGF-C	Tetraacetylated N-azidoacetyl-d-mannosamine-loaded nanoparticles target and tag with azides the PDAC cells and their EVs [218]. Modified nanoparticles (dibenzyl-cyclootyne) inhibit the proliferation of PDAC cells and EV secretion [218]. Suppression of EV-VEGF-C expression by selective HDAC1/2 inhibitor (B390), and suppression of lymphangiogenesis and lymphatic tumor dissemination [219].
	Exosome inhibitor GW4869	EV-mir-21	PTEN expression restored in vivo and in vitro suppression of PDAC progression and chemoresistance [220].
	Inhibition of CD9-positive ANXA6-EVs	ANXA6-EVs	Reduced the stromal modification and tumor progression [140].
Photodynamic therapy and immunotherapy	clorin e6	PDAC cells	PDAC-derived CeR-R exo increases the photosensitivity of PDAC [221]. Stimulatory effect on immune cells, via inducing cytokine release by immune cells. Increase in reactive oxygen species in PDAC cells, under the effect of the irradiation laser [220].

Engineered EVs (iExosomes); mesenchymal stromal cells of the human umbilical cord (UC-MSC-derived EVs); bone marrow mesenchymal stem cell (BM-MSC); oxaliplatin (OXA); microenvironment (TME); re-assembled exosomes (R-Exo); dual specificity phosphatase-2 (DUSP2).

## Data Availability

Not applicable.

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
