# Peer review of "A Current Synopsis of the Emerging Role of Extracellular Vesicles and Micro-RNAs in Pancreatic Cancer: A Forward-Looking Plan for Diagnosis and Treatment"

_ijms, 2024, doi:10.3390/ijms25063406_

Round 1

Reviewer 1 Report

Comments and Suggestions for Authors

A very detailed, well-written manuscript on the role of extracellular vesicles and in particular microRNAs exposed by EVs in pancreatic cancer. The Greek-American group of scientists prepared this manuscript extremely carefully using many references. The manuscript is enriched with numerous tables and figures that facilitate understanding of a topic that is important both from the scientific and clinical point of view, in addition, the manuscript can serve as a great source of information for students and scientists starting their scientific adventure with EVs.

I do not have any major comments on the manuscript because it has been prepared very carefully, but I am asking the authors to make minor corrections, namely:

1. I would like to ask you to remove the dot from the title,

2. I would like to ask the authors to expand the conclusions,

3. I would like to ask you to expand the references to include the following manuscripts:

https://www.mdpi.com/2073-4409/11/18/2913

https://www.mdpi.com/2072-6694/14/14/3552

https://www.mdpi.com/2072-6694/15/7/2158

https://www.mdpi.com/1422-0067/21/15/5373

https://www.mdpi.com/1422-0067/21/17/6097

Author Response

Reply: Firstly, the authors would like to thank Reviewer #1 for his/her constructive suggestions for our manuscript. Please see below for detailed responses to his/her minor/major comments.

  1. We removed the dot from the title
  2. We expanded the conclusions
  3. We added in our references the manuscripts that you suggested. ( updated  references 23,108 ,226,

Reviewer 2 Report

Comments and Suggestions for Authors

Dear authors
You wrote a very lengthy and meticulous review on exosomes and their load for an earlier diagnosis of pancreatic ductal adenocarcinoma. Scientifically your paper is accurate and has a huge amount of data.

The main objection to the paper is that it contains four different topics and therefore it is not focused on what is supposed to be the aim of your work. Actually these are four different reviews in one. In a book they would represent four different chapters. That is fine for a book. For a review article, it is not fine. Furthermore, it seems that these four different chapters were written by different persons. That is also fine. What is not fine when you incur in repetitions. More details below.

Your literary approach to this complex subject is disorganized and this makes reading difficult. More details below.
I think you will have to rewrite the paper with a more organized, unified and focused approach.
Think of who will be the reader.
If the reader is a non-expert it is acceptable to give a textbook description of exosome biogenesis and what is a non-coding RNA, as you did. But someone who is not in this field will not read this type of paper. Furthermore, the detailed and lengthy description of exosome biogenesis and non-coding RNA makes the reader`s mind loose the focus on the early diagnostic purpose which is the supposed scope of the paper.

If the reader is an expert, these two first chapters are absolutely unnecessary and only add to a heavy burden accompanied with poor figures.
I simply think that the first two "chapters" considered in the paper should be eliminated or drastically shortened as an introduction. The basic textbook knowledge of exosomal biogenesis and non-coding RNA regulation are supposedly known issues by someone who reads this kind of articles. Otherwise, on the same token you should explain also what is a cell or what is cancer.

Now, lets get into details.

UNFOCUSED: it takes you six pages of introductory (and elementary) descriptions to arrive to the main core of your review. This is a little bit too much. I suggest you fully delete point 2. At the most, you should refer the reader to other reviews or books on EV and miRNAs biogenesis. There are excellent and focusedpublications on these subjects.
E.g.
Hessvik NP, Llorente A. Current knowledge on exosome biogenesis and release. Cellular and Molecular Life Sciences. 2018 Jan;75:193-208. 
Zhang Y, Liu Y, Liu H, Tang WH. Exosomes: biogenesis, biologic function and clinical potential. Cell & bioscience. 2019 Dec;9(1):1-8.
Han QF, Li WJ, Hu KS, Gao J, Zhai WL, Yang JH, Zhang SJ. Exosome biogenesis: Machinery, regulation, and therapeutic implications in cancer. Molecular Cancer. 2022 Dec;21(1):1-26.
REPETITIONS: In point 4 initial sentences, you repeat the same concepts that you have already said in the introduction. There is no need for a second introduction of the same sort.

Table 1 is a repetition of the same concepts and data you established in the previous text. Whether you keep the table or the text, both are too much.

Same thing happens with Table 2.

DISORGANIZED
You discuss all miRNAs intermingled. This makes reading heavy and difficult.
This is what you wrote: One of the major oncomiRs that is highly expressed in serum or tissue biopsies, is 253 miR-21. MiR-21 expression levels are closely related to the regulation of tumor-suppressor 254 genes that are implicated in pivotal cell functions and pathways, such as apoptosis and 255 Ras-Raf-MEK-ERK pathways or/and epidermal growth factor receptor (EGFR), or/and 256 PI3K/AKT signaling pathways, respectively [36,37]. More particularly, miR-21 overex- 257 Int. J. Mol. Sci. 2024, 25, x FOR PEER REVIEW 7 of
49 pression induces the growth and proliferation of pancreatic cancer cells, while it concom- 258 itantly inhibits their apoptosis, resulting in a decontrolled cell cycle [36,37]. The levels of 259 MiR-186 are also highly found in PDAC, which also induces proliferation and metastasis 260 via targeting Nuclear Receptor Subfamily 5 Group A Member 2 (NR5A2) gene that en- 261 codes the transcription factor NR5A2, leading to several deregulations in gene expression 262 [38]. Likewise, miR-17-5p overexpressed levels are closely implicated in the cell cycle de- 263 regulation, via interrupting the expression of RBL2/E2F4 repressing complexes [39], while 264 miR-196b suppresses the apoptotic mechanism via targeting CADM1[40]. Meanwhile, 265 miR-18a, which is a member of the oncogenic miR-17-92 cluster, is highly expressed in 266 PDAC, whereas its levels are significantly reduced after surgical treatment. This miRNA 267 is highly
implicated in proliferation and MYC-induced transcriptional activation [41]. An- 268 other oncomir that is identified in PDAC cases is miR-191, which is implicated in the mod- 269 ification of the extracellular matrix and the promotion of distant tumor cell dissemination 270 [42], while miR-29a and miR-221 are also implicated in PDAC progression, promoting 271 invasion and metastasis [43-45]. 272 Furthermore, Mir-301a- 3p and miR-374 are also having an oncogenic role in PDAC 273 via inducing migration and increased invansiveness of pancreatic cancer cells, with the 274 former targeting SMAD4 [46], whereas the latter via deregulating Secernin 1 (SRCIN1), 275 expression, leading to its low expression, leading to EMT and PDAC progression [47]. 276 
Meanwhile, miR-1469-5p is correlated to the overproliferation of PDAC cells, via interact- 277 ing with NDRG1/NF-κB/E-cadherin pathway[48], while miR-205 is implicated with the 278 Wnt signaling pathway, increasing the proliferation via targeting the suppressive gene 279 that encodes Adenomatous polyposis coli (APC) [49]. Last but not least, miR-10b overexpres- 280 sion in PDAC is closely involved with tumor invasive
behavior and progression via in- 281 hibiting TIP30 expression and promoting EGF and TGF-β effects, leading to a generally 282 worrisome prognosis [50]. In Table 1 we demonstrate the oncogenic effect of several miR- 283 NAs in PDAC. 284 Table 1. OncomiRs in PDAC.
This is how it should be:
â–º miR-21: One of the major oncomiRs that is highly expressed in serum or tissue biopsies, is miR-21. MiR-21 expression levels are closely related to the regulation of tumor-suppressor genes that are implicated in pivotal cell functions and pathways, such as apoptosis and Ras-Raf-MEK-ERK pathways or/and epidermal growth factor receptor (EGFR), or/and PI3K/AKT signaling pathways, respectively [36,37]. More particularly, miR-21 overexpression induces the growth and proliferation of pancreatic cancer cells, while it concomitantly inhibits their apoptosis, resulting in a decontrolled cell cycle [36,37].
â–º miR 186: The levels of MiR-186 are also highly found in PDAC, which also induces proliferation and metastasis via targeting Nuclear Receptor Subfamily 5 Group A Member 2 (NR5A2) gene that encodes the transcription factor NR5A2, leading to several deregulations in gene expression 262 [38].
â–º miR-17-5p overexpressed levels are closely implicated in the cell cycle de-
regulation, via interrupting the expression of RBL2/E2F4 repressing complexes [39],
â–º miR-196b suppresses the apoptotic mechanism via targeting CADM1[40].
â–º miR-18a, which is a member of the oncogenic miR-17-92 cluster, is highly expressed in PDAC, whereas its levels are significantly reduced after surgical treatment.
â–º miRNA 267 is highly implicated in proliferation and MYC-induced transcriptional activation [41].
â–º miR-191, is implicated in the modification of the extracellular matrix and the promotion of distant tumor cell dissemination [42],
â–º miR-29a and miR-221 are also implicated in PDAC progression, promoting invasion and metastasis [43-45].
â–º Mir-301a-3p and miR-374 are also having an oncogenic role in PDAC via inducing migration and increased invasiveness of pancreatic cancer cells, with the former targeting SMAD4 [46], whereas the latter via deregulating Secernin 1 (SRCIN1), expression, leading to its low expression, and to EMT and PDAC progression [47].

â–º miR-1469-5p is correlated to the overproliferation of PDAC cells, via interacting with NDRG1/NF-κB/E-cadherin pathway[48].
â–º miR-205 is implicated with the Wnt signaling pathway, increasing the proliferation via targeting the suppressive gene that encodes Adenomatous polyposis coli (APC) [49].
â–º miR-10b overexpression in PDAC is closely involved with tumor invasive behavior and progression via inhibiting TIP30 expression and promoting EGF and TGF-β effects, leading to a generally poor prognosis [50].
In Table 1 we demonstrate the oncogenic effect of several miR- 283 NAs in PDAC. Comment. in Table I you do not demonstrate anything. You only repeat what you said above.
ENGLISH SYNTAX
In general your syntax is sort of poor. I am not a native English speaker, therefore I am not entitled to correct your English. You should have the paper corrected by a native English speaker.
In spite of all these critics, I think that your paper is worth to be reformatted because it deserves to be published.

Comments on the Quality of English Language

It is bad English. Should be corrected by a native English speaker

Author Response

Firstly, the authors would like to thank Reviewer #2 for his/her constructive suggestions for our manuscript. Please see below for detailed responses to his/her minor/major comments. The manuscript was written and corrected by a native speaker.

The special issue is about exosomes and non-coding RNAs.  Indeed, this is a large review, however, we believe that the reader could have a nice overview of these subjects, which are connected based on intercellular communication. This review aims to sum up the current information about this field, in the most “concentrated” way, however in a way that the reader could have an overview of the biogenesis and pathophysiology.

Even though this review is long, we believe that the sequence of the given information is logical and the titles are explainable. When we write a manuscript we don’t focus only on the field of the experts, but also on people who are new in the field and try to understand what are these nanoparticles and how they can be utilized as diagnostic and therapeutic tools.

The focus was not only on the early diagnosis.  The aim was to make an overview of all the current papers and data about the utilization of micro-RNAs and EVs as therapeutic and diagnostic tools.

For the readers who are more expert the part of biogenesis is considered adequate, for the readers, who are new in the field could be considered quite detailed. However, this issue remains controversial for all the published manuscripts. We don’t want our paper to be limited only to the scientists who are interested in this field. That’s why we believe that the biogenesis of EVs and miRNAs should remain, as they explain the reasons why these molecules can be utilized as therapeutic and diagnostic tools.  The figures have been made by Biorender, which is utilized by a wide variety of well-known universities. The details that are shown in these figures help the reader to understand the biogenesis of miRNAs and EVs. 

Additionally, We added some of the suggestions to our reference list (updated 19) and part 4 was reduced. 

The tables aim to summarize the content of the paragraphs and make the information more “approachable” for the reader. We believe that their presence is helpful, however, we reduced the information that is displayed on them (also in Table 3) . The reason that we provide tables is that we want the reader to quickly see the table and have an overview of what is written in the paragraph.

Furthermore, based on the Template of IJMS we didn’t see any form of writing like this, that’s why we wrote it down as a paragraph. However, if the Editor allows it we can proceed to the modification of the format.

As we responded above, the table aims to summarize the content of the paragraphs and make the information more “approachable” for the reader. The reader can quickly see the table and have an overview of what is written in the paragraph.

Finally, the manuscript is corrected by a native speaker, as well as by the authors who wave high-level English language certificates.  Thank you for your time to revise our manuscript.

Reviewer 3 Report

Comments and Suggestions for Authors

The manuscript titled "A current synopsis of the emerging role of extracellular vesicles and micro-RNAs in pancreatic cancer: A forward-looking plan for diagnosis and treatment" aims to provide a comprehensive review of the current knowledge regarding the role of extracellular vesicles (EVs) and micro-RNAs (miRNAs) in the suppression or progression of pancreatic cancer. The authors also aim to provide evidence supporting the use of EVs and miRNAs as potential biomarkers for earlier diagnosis of pancreatic cancer in patients. The authors do an excellent job of covering these topics in depth. The manuscript is very detailed, outlining the mechanisms for EV and miRNA generation, which EVs and miRNAs have been measured in different tissues, and the specific effects of each individual EV and miRNA in pancreatic cancer. The authors also report how these potential biomarkers can be used to diagnose pancreatic cancer in patients separately from other cancers and inflammatory diseases. The figures and tables efficiently illustrate mechanisms and allow the reader to quickly identify the individual effects of various biomarkers on the progression of pancreatic cancer. The manuscript is acceptable in its current form.

Author Response

The authors would like to thank Reviewer #3 for his/her time to revise this manuscript and for his/her comments.

Reviewer 4 Report

Comments and Suggestions for Authors

Pancreatic ductal adenocarcinoma (PDAC) comprises one of these highly deadly malignancies worldwide. The discovery of novel diagnostic and therapeutic tools is considered a necessity for this tumor, due to its low survival rates and treatment failures. One of the extensively investigated potential diagnostic and therapeutic modalities includes extracellular vesicle (EVs). The authors reviewed the last researches of PDAC-related EVs, and discussed the role of EVs and miRNAs in pancreatic cancer, as well as their potent utilization as diagnostic and therapeutic tools. The topic is interesting, and the manuscript is well written. After some minor revision could be suitable for publication.

l   Line 260. “MiRNA-186” should be “miRNA-186”.

l   Table 1, 2, 3, 4 and 5. Please unify the format. Please use “miR-xxx” in the table.

Author Response

 Reply: Firstly, the authors would like to thank Reviewer #4 for his/her constructive suggestions for our manuscript. Please see below for detailed responses to his/her minor comments.

  • Line 260. “MiRNA-186” should be “miRNA-186”.

Reply:  Thank you for your correction. We modified it as you suggested

  • Table 1, 2, 3, 4 and 5. Please unify the format. Please use “miR-xxx” in the table.

Reply: Thank you for your correction. We modified the format.

Round 2

Reviewer 1 Report

Comments and Suggestions for Authors

I still rate the manuscript very highly, although, after the revision, the identity coefficient with other publications increased significantly.

I ask the authors to revise the literature I originally recommended, but it seems to me that not all manuscripts have been included.

Then please check whether the numbering of the papers in the literature list has not changed.

Moreover, it is a very well-prepared manuscript, congratulations to the authors once again.

Author Response

Reply: Firstly, the authors would like to thank Reviewer #1 once again for his/her constructive suggestions for our manuscript.

We revised the skeleton of this manuscript to be clear for the reader, as well as we re-vised the English once again, as we observed some grammar and syntactic errors. As we previously responded, we included the majority of the recommended articles in our manuscript, based on their content. The updated references of the manuscripts that you suggested are number 23, 108, and 226. The numbering of the papers in the list has not been changed because we included your suggestions instead of other papers that were previously included (they were older than the ones you suggested).

We thank you once again for your precious suggestions, which increased the scientific impact of our manuscript.

Reviewer 2 Report

Comments and Suggestions for Authors

I found no substantial changes in the new revised manuscript.

I think it is difficult to read and the figures are very poor. Bio render is no warrant of quality.

I can only repeat all the observations I made in the first revision.

Comments on the Quality of English Language

No comments

Author Response

Reply: Firstly, the authors would like to thank Reviewer #2 once again for his/her constructive suggestions for our manuscript.

  1. We significantly reduced the amount of complex sentences
  2. We revised the manuscript and corrected all the remaining grammar and syntactic errors
  3. We significantly changed the skeleton of the manuscript and the form of the paragraphs exactly as you suggested and we believe that your suggestion was precious.
  4. The content in this revised form is clear and easily read
  5. We increased the size of the figures. We agree that Biorender doesn’t guarantee quality. We mentioned that platform due to the quality of the icons that are provided and its global utilization by many well-known universities and high-impact journals. We still consider the figures helpful for young scientists to understand the mechanisms of biogenesis of EVs and miRNAs, as well as their utilization as therapeutic and diagnostic tools.

If we delete the figures, the less-expert readers will have a hard time understanding the procedure without a schematic presentation.

Based on several papers that we revised (also from journals of high impact), we believe that we included in these figures the most important molecules and mechanisms.

We thank you once again for your precious suggestions, which significantly increased the scientific impact of our manuscript.

Round 3

Reviewer 2 Report

Comments and Suggestions for Authors

You have made a huge effort to improve the organization and readibility of the paper. Now, it seems much easier to read whether for an expert or a junior scientist.

Some more suggestions

The figures are a problem. A simple way to improve figure 1 is by removing the left upper corner panel of apoptosis, which has no sense to be shown, and substantially enlarge the rest of the figure. The present size is inappropriate for understanding exosome's biogenesis. 

Figure 2 is also very difficult to understand for the junior reader. I would replace that figure for something that is not so artistic, but much easier to understand such as the figure I inserted here.

You can freely use the figure. It has no copyright.

There are some typos.

Comments on the Quality of English Language

Acceptable

Author Response

Reply: 

All the authors thank you for your precious suggestions. 

  1. We made the modification you suggested in Figure 1. More specifically we removed the part of apoptotic bodies and we modified the dimensions
  2. We re-made the figure 2 less artistic and more easily explained

(Ps we did not find any inserted figure).

  1. We re-corrected the typos and we re-checked the manuscript for syntactic and grammar errors.

Thank you for your substantial contribution that improved the scientific impact of this manuscript

We hope that the new forms of our figures are acceptable.